# Polar Lows – Moist Baroclinic Cyclones Developing in Four Different Vertical Wind Shear Environments

Patrick Johannes Stoll[1], Thomas Spengler[2], Annick Terpstra[2], and Rune Grand Graversen[1,3]

[1]Department of Physics and Technology, Arctic University of Norway, Tromsø, Norway
[2]Geophysical Institute, University of Bergen, and Bjerknes Centre for Climate Research, Bergen, Norway
[3]Norwegian Meteorological Institute, Norway

**Correspondence:** Patrick Johannes Stoll (patrick.stoll@uit.no)

**Abstract.** Polar lows are intense mesoscale cyclones that develop in polar marine air masses. Motivated by the large variety of their proposed intensification mechanisms, cloud structure, and ambient sub-synoptic environment, we use self-organising maps to classify polar lows. The method is applied to 370 polar lows in the North-East Atlantic, which were obtained by matching mesoscale cyclones from the ERA-5 reanalysis to polar lows registered in the STARS dataset by the Norwegian Meteorological Institute. ERA-5 reproduces most of the STARS polar lows.

We identify five different polar-low configurations which are characterised by the vertical wind shear vector, the change of the horizontal-wind vector with height, relative to the propagation direction. Four categories feature a strong shear with different orientations of the shear vector, whereas the fifth category contains conditions with weak shear. This confirms the relevance of a previously identified categorisation into forward and reverse-shear polar lows. We expand the categorisation with right and left-shear polar lows that propagate towards colder and warmer environments, respectively.

For the strong-shear categories, the shear vector organises the moist-baroclinic dynamics of the systems. This is apparent in the low-pressure anomaly tilting with height against the shear vector, and the main updrafts occurring along the warm front located in the forward-left direction relative to the shear vector. These main updrafts contribute to the intensification through latent-heat release and are typically associated with comma-shaped clouds.

Polar low situations with a weak shear, that often feature spirali-form clouds, occur mainly at decaying stages of the development. We thus find no evidence for hurricane-like intensification of polar lows and propose instead that spirali-form clouds are associated with a warm seclusion process.

## 1 Introduction

Polar lows (PLs) are intense mesoscale cyclones with a typical diameter of $200 \text{-} 500 \, \text{km}$ and a short lifetime of $6 \text{-} 36 \, \text{h}$ that develop in marine cold-air outbreaks during the extended winter season (e.g Rasmussen and Turner, 2003; Yanase et al., 2004; Claud et al., 2004; Renfrew, 2015; Rojo et al., 2015). Despite numerical weather-prediction models being capable of simulating these systems, operational forecasts still have have issues predicting the exact location and intensity of PLs (e.g. Kristjánsson et al., 2011; Føre et al., 2012; Stoll et al., 2020). Several paradigms have been proposed to describe PL development, ranging from baroclinic instability (e.g. Harrold and Browning, 1969; Reed, 1979; Reed and Duncan, 1987) to symmetric hurricane-like

growth (e.g. Rasmussen, 1979; Emanuel and Rotunno, 1989). The multitude of paradigms demonstrates that our dynamical interpretation of these systems is still deficient (Jonassen et al., 2020). To further alleviate this shortcoming, we present a classification of PLs by their structure and sub-synoptic environment to identify the relevance of the proposed paradigms.

The proposed PL paradigms often stem from the different cloud structures (Forbes and Lottes, 1985; Rojo et al., 2015) and sub-synoptic environments (Duncan, 1978; Terpstra et al., 2016). From the 1980s, the PL spectrum was thought to range

from pure baroclinic PLs with a comma-shaped cloud structure to convective systems with a spirali-form cloud signature like hurricanes (p.157 Rasmussen and Turner, 2003). The latter types usually were argued to be invoked either conditional instability of the second kind (CISK Charney and Eliassen, 1964; Ooyama, 1964) or by wind-induced surface heat exchange (WISHE Emanuel, 1986). However, most PLs appear as hybrids between the extremes of this spectrum (Bracegirdle and Gray, 2008).

Using idealised simulations to map the sensitivities of cyclone development in the PL spectrum, Yanase and Niino (2007) found that cyclones intensify fastest in an environment with high baroclincity where latent heat release supports the development. As neither dry baroclinic nor pure CISK modes grow fast enough to explain the rapid intensification of PLs, the growth is most likely associated with moist baroclinic instability (Sardie and Warner, 1983; Terpstra et al., 2015).

Furthermore, hurricane-like PLs rarely resemble the structure of hurricanes; instead they feature asymmetric updrafts typical

of baroclinic development with latent heating not playing the dominant role (Føre et al., 2012; Kolstad et al., 2016; Kolstad and Bracegirdle, 2017). The PLs that appear hurricane-like in their mature stage appear to be initiated by baroclinic instability (e.g. Nordeng and Rasmussen, 1992; Føre et al., 2012).

Several categorisations of PLs were proposed to shed light on the development pathways of PLs (e.g. Duncan, 1978; Businger and Reed, 1989; Rasmussen and Turner, 2003; Bracegirdle and Gray, 2008; Terpstra et al., 2016). Duncan (1978) suggested

a categorisation based on the vertical wind-shear angle of the PL environment, defined as the angle between the tropospheric mean wind vector and the thermal wind vector. Situations where the vectors point in the same (opposite) direction are referred to as forward (reverse) shear conditions. PLs have been found to occur in both forward-shear (e.g. Reed and Blier, 1986; Hewson et al., 2000) and reverse-shear environments (e.g. Reed, 1979; Bond and Shapiro, 1991; Nordeng and Rasmussen, 1992), where both types of PLs occur approximately equally often (Terpstra et al., 2016).

PLs in forward-shear environments develop similar to typical mid-latitude cyclones in a deep-baroclinic zone with an associated upper-level jet. They have the cold air to the left with respect to their direction of propagation and are mainly propagating eastward (Terpstra et al., 2016). PLs in reverse-shear environments, on the other hand, often develop in the vicinity of an occluded low-pressure system and are characterised by a low-level jet. They have the cold air to the right with respect to their propagation direction and are mainly propagating southward. Reverse-shear PLs also feature considerably higher surface heat

fluxes and a lower static stability in the troposphere, expressed by a larger temperature contrast between the temperature at the sea surface and at $500\,\mathrm{hPa}$ (Terpstra et al., 2016).

However, assigning PL environments solely based on two types of shear conditions might be insufficient to characterise all PLs types. The shear angle cannot distinguish between baroclinically- and convectively-driven systems, hence it cannot address the hurricane-like part of the PL spectrum. To alleviate this shortcoming, we categorise PLs based on their sub-synoptic

environment using self-organising maps (SOM, see Section 2.5). The thereby identified meteorological configurations reveal the underlying PL intensification mechanisms, allowing us to investigate the following research questions:

– What are the archetypal meteorological conditions during PL development?

– Can the existing PL classification be confirmed or should it be revised?

– What are the pertinent intensification mechanisms?

## 2 Methods

### 2.1 Polar-low list

This study is based on the Rojo list (Rojo et al., 2019), a modified version of the STARS (Sea Surface Temperature and Altimeter Synergy for Improved Forecasting of Polar Lows) dataset. This list includes the location and time of PLs detected from AVHRR satellite images over the North-East Atlantic that were listed in the STARS dataset by the Norwegian Meteorological Institute between November 1999 and March 2019 (Noer and Lien, 2010). The STARS dataset has been used for several previous PL studies (e.g. Laffineur et al., 2014; Zappa et al., 2014; Rojo et al., 2015; Terpstra et al., 2016; Smirnova and Golubkin, 2017; Stoll et al., 2018).

The advantage of the Rojo list compared to the STARS dataset is that it contains considerably more PL cases. While the STARS dataset only includes the major PL centre for events of multiple PL developments, the Rojo list includes the location of all individual PL centres (Rojo et al., 2015). Thus, the Rojo list includes 420 PL centres, which are associated with the 262 PL events in the STARS database of which 183 PL events feature a single PL centre and the remaining 79 events have 2 - 4 PL centres per event. In addition, the Rojo list classifies the cloud morphology for each detected PL time step.

As the Rojo list includes individual time steps of each PL when AVHRR satellite images were available, the time interval between observations is irregular and ranges from 30 minutes and up to 12 hours. This also implies that the list in many cases lacks the genesis and lysis time of some PLs.

### 2.2 Polar-low tracks in ERA-5

We use the European Centre for Medium-Range Weather Forecasts (ECMWF) state-of-the-art reanalysis version 5 (ERA-5 Hersbach and Dee, 2016) to track PLs and analyse the atmospheric environment. The ability of ERA-5 to simulate PLs has not yet been investigated, though some studies have shown that atmospheric models with a comparable horizontal resolution to ERA-5 are capable to produce most PL cases (Smirnova and Golubkin, 2017; Stoll et al., 2018). Further, Stoll et al. (2020) demonstrated that the ECMWF model at comparable resolution as ERA-5 reproduced the 4 dimensional structure of a particular PL case reasonably well. As other studies estimated the amount of represented STARS PLs in ERA-Interim, the precursor reanalysis to ERA-5, to be 48% Smirnova and Golubkin (2017), 55% (Zappa et al., 2014), 60% (Michel et al., 2018), and 69%

(Stoll et al., 2018), we anticipate the performance of ERA5 to be even higher. Note that this study does not rely on a realistic

reproduction of the PLs, as we mainly focus on the PL environment that has a spatial scale that is well captured by ERA-5.

The ERA-5 model provides data from 1950 to the near-present at hourly resolution with a spectral truncation of T639 in the horizontal direction, which is equivalent to a grid spacing of about $30\,\mathrm{km}$. The reanalysis has 137 hybrid levels in the vertical, of which approximately 47 are below $400\,\mathrm{hPa}$, which is the typical height of the tropopause in polar-low environments (Stoll et al., 2018). We obtained the data at a lat $\times$ lon grid of $0.25° \times 0.5°$ within $50° \text{-} 85°\mathrm{N}$ and $40°\mathrm{W} \text{-} 65°\mathrm{E}$. We chose a coarser

grid spacing for the longitudes due to their convergence towards the pole.

To analyse the PL development, we derive PL tracks with an hourly resolution by applying the mesoscale tracking algorithm developed by Watanabe et al. (2016) and retain the tracks that match the Rojo list. The tracking procedure is based on detecting local maxima in relative vorticity at $850\,\mathrm{hPa}$, where we tuned the parameters in the tracking algorithm based on the objective to obtain a good match with the Rojo list. In particular, we modified the algorithm as follows:

– No land mask is applied in order to include PLs propagating partially over land.

– A uniform filter with a $60\,\mathrm{km}$ radius is applied to the relative vorticity to reduce artificial splitting of PL tracks and to smooth the tracks.

– Parameters in the algorithm by Watanabe et al. (2016) are adapted; the vortex peak to $\zeta_{max,0}$ is $1.5 \times 10^{-4}\,\mathrm{s}^{-1}$, and the vortex area to $\zeta_{min,0}$ is $1.2 \times 10^{-4}\,\mathrm{s}^{-1}$.

– We do not require a filter for synoptic-scale disturbances, as the matching to the Rojo list ensures that the tracks include PLs only.

All tracks that have a distance of less than $150\,\mathrm{km}$ to a PL from the Rojo list for at least one time step are regarded as matches. In total we obtain 556 associated PL tracks. Multiple matches to the same PL from the Rojo list occur due to two reasons: Firstly, ERA-5 features multiple PLs connected to one PL centre from the Rojo list. Secondly, the tracking algorithm

yields several track segments for the same PL. The latter occurs if the location of the vorticity maximum moves within an area of high vorticity, e.g., a frontal zone. In these cases, two track segments are merged provided that the time gap between them is less than 6 hours, and that the extrapolation of one track segment over the time gap includes the other segment within a distance of $150\,\mathrm{km}$. This merging was applied for 86 PL tracks, such that 470 PL tracks remain.

We exclude tracks having a lifetime shorter than $5\,\mathrm{hours}$ (54 tracks) and being located over land for most of the PL lifetime

(5 tracks). The latter is defined as when the initial, middle, and final time step of the PL all occur on land. Other land exclusion methods were tested and gave similar results. If a track is included twice, which occurs when it matches with two PLs from the Rojo list, one is removed (37 tracks), and is labelled as a match to both PLs from the Rojo list.

Applying this procedure, a total of 374 out of the initial 556 PL tracks is retained with 13,221 hourly time steps. The list with the obtained PL tracks is provided in the supplement. We compared a random subset of obtained tracks with the PLs from

the Rojo list, satellite imagery, and ERA-5 fields and concluded that the vast majority of the obtained tracks can be considered to be PLs.

The resulting mean lifetime of ∼35 h (13,221 hourly time steps divided by 374 PLs) is slightly longer than that observed by Rojo et al. (2015), who found that two thirds of the PLs detected from satellite images lasted for less than 24 h. This difference may be explained by biases in both the model and the observational dataset. Our reanalysis-based dataset is likely biased towards longer lifetimes, as longer-living and larger PLs are better reproduced by the reanalysis. In contrast, satellite-based datasets, such as Rojo et al. (2015), are likely underestimating the lifetime based on the first (last) satellite image from which a PL may be observed after (before) the PL genesis (lysis) for two reasons: (i) Sometimes the time gap between two consecutive satellite images is multiple hours (Rojo et al., 2015), (ii) the PL is not identified due to other disturbing cloud structures (e.g. Furevik et al., 2015).

The number of detected PL tracks (374) is close to the number of PL centres included in the Rojo list (420). The 374 PL tracks are associated with 244 different PL events from the Rojo list, implying that 244 of the 262 PL events (93%) from the Rojo list are reproduced in ERA-5. These 374 PL tracks are associated with 348 different PL centres from the Rojo list. As mentioned earlier, sometimes multiple PLs are observed in ERA-5 associated to one PL from the Rojo list, which appears realistic after investigating these cases, as also the Rojo list misses some PL centres.

In total, for 348 of the 420 centres (83%) from the Rojo list at least one associated PL is found. In the Norwegian Sea, 219 of 255 (86%) PL centres from the Rojo list are reproduced, whereas 129 of 165 (78%) are detected in the Barents Sea, where PLs in the Norwegian and Barents Sea are separated by the longitude of the first time step being smaller or larger than 20° E, respectively. A higher detection rate of STARS PLs for the Norwegian than for the Barents Seas was also observed by Smirnova and Golubkin (2017). It may be explained by STARS PLs being larger in the former than in the latter ocean basin (Rojo et al., 2015) and larger systems being more likely captured by ERA-5. The high detection rates indicate the capability of ERA-5 to represent most PLs and highlights that ERA-5 is superior to its predecessor ERA-Interim when it comes to capturing PLs, where Stoll et al. (2018) detected 55% and Michel et al. (2018) about 60% of the STARS PLs in ERA-Interim. Note that the detection rates depend on the applied matching criteria. The matching criteria applied here are stronger than those applied in Stoll et al. (2018).

## 2.3 Polar-low centred analysis

We employ a PL-centred analysis, where meteorological fields from ERA-5 for each individual time step of a PL track are transformed onto a PL-centred grid. The cells of the PL-centred grid (black dots in Figure 1) are derived from the location and propagation direction of the PL. The propagation direction is obtained by applying a cubic-spline smoothing to the PL track points (De Boor, 1978). We smooth, because the tracks can feature non-monotonous behaviour due to the discrete nature of the grid and varying locations of the vorticity maxima in areas of enhanced vorticity, such as frontal zones. Smoothing also provides continuous tracks in situations of low propagation speed, where the propagation direction is highly variable.

Meteorological fields from ERA-5 were linearly interpolated onto the PL-centred grid with an extent of $1000 \times 1000\,\text{km}$, and with a grid spacing of 25 km (see Fig. 1). Given a typical PL diameter between 150 and 600 km (Rojo et al., 2015), this grid covers the PL and its sub-synoptic environment.

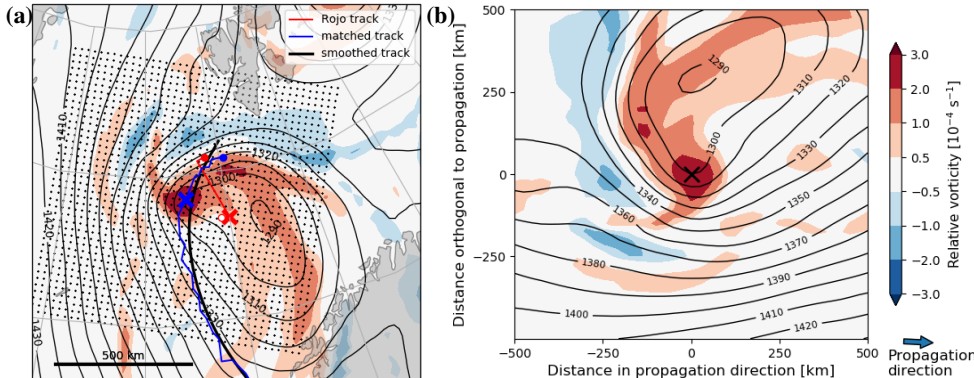

**Figure 1.** Exemplary depiction of deriving the PL-centred analysis. (a) Relative vorticity at 850 hPa (shading) and geopotential height at 850 hPa (contours, spacing 10 m) for 20 March 2001 01:00 UTC. The track of polar low number 10 from the Rojo list is shown in red and the matched track is depicted in blue. The location at the selected time are indicated by 'x'. The smoothed track is depicted in black, the propagation direction of the PL at the selected time is southward. (b) Same fields as for (a) in the polar-low centred domain, constructed such that the polar-low centre is in the middle ('x') and the propagation direction is towards the right.

We exclude time steps where this grid is not fully within the chosen ERA-5 boundaries (Section 2.2). This reduces the PL time steps from 13,221 to 12,695 and removes 4 of the 374 PLs for their complete lifetime. As most of the excluded time steps occur at the end of the PL lifetime, this exclusion has insignificant influence on our analysis of the PL intensification.

## 2.4 Parameter preparation

In addition to standard meteorological variables, we derived an additional set of parameters using grid points within a radius of 250 km around the PL centre. For near-surface parameters, such as the 10 m wind vector, the turbulent heat fluxes, and parameters partly derived from low-levels, such as the Brunt-Väisälä frequency, only grid cells over open ocean, defined by a water fraction larger than 80%, are included. Presented results were found qualitatively insensitive to variations in the radius.

We estimate the tropospheric **Brunt-Väisälä frequency**, $N$, using the potential temperature, $\theta$, at 500 and 925 hPa

$$N = \sqrt{\frac{g}{\overline{\theta}}\frac{\partial \theta}{\partial z}} \approx \sqrt{\frac{g}{(\theta_{500} + \theta_{925})/2}\frac{\theta_{500} - \theta_{925}}{z_{500} - z_{925}}} \tag{1}$$

with gravitational constant $g = 9.8\,\mathrm{ms^{-2}}$ and geopotential height $z$. Afterwards, we apply a radial average to obtain the environmental value.

The **differential wind vector** is computed using the mean wind vectors at 500 and 925 hPa,

$$\Delta\overline{\boldsymbol{u}} = (\Delta\overline{u}, \Delta\overline{v}) = (\overline{u_{500}} - \overline{u_{925}}, \overline{v_{500}} - \overline{v_{925}}), \tag{2}$$

where an overbar indicates the mean computed within a radius of 500 km around the PL centre. The upper-level (500 hPa) is considerably higher than the level (700 hPa) chosen by Terpstra et al. (2016). Given that PLs span the entire depth of the polar

troposphere, we chose one level from the lower and one from the upper troposphere, with neither level intersecting with the sea surface nor the tropopause.

We define the **vertical-shear strength** as

$$\left| \frac{\Delta \overline{u}}{\Delta \overline{z}} \right| = \frac{|\Delta \overline{u}|}{\overline{z}_{500} - \overline{z}_{925}}. \tag{3}$$

The **vertical-shear angle**, $\alpha = [\alpha_s - \alpha_p] \,(\mathrm{mod}\, 360°)$, is derived from the angle between the differential wind vector, $\Delta \overline{u}$, and the propagation direction, $\alpha_p$, of the PL. The propagation vector of the PL appears to be a good estimate of the mean wind encountered by the PL (Fig. 3). Different to Terpstra et al. (2016), we compute the radial average before calculating the angle. Thereby, our method obtains the shear angle of the environmental mean flow and partially filters for perturbations induced by the PL itself. Also to reduce the influence of the PL-induced perturbations, a large radius of 500 km is applied for the

computation of the differential wind vector. Note, that the shear angle by this computation takes values between 0 and 360° and not between 0 and 180° as in Terpstra et al. (2016). In addition, we define the **vertical-shear vector** in the propagation direction

$$\left( \frac{\Delta \overline{u}}{\Delta \overline{z}} \right)_p = \left| \frac{\Delta \overline{u}}{\Delta \overline{z}} \right| \times (\cos(\alpha), \sin(\alpha)). \tag{4}$$

Our method is different to Duncan (1978) and Terpstra et al. (2016), though tests confirm that their thermal wind vector is

qualitatively similar to the vertical-shear vector utilised in this study.

The **vorticity tendency** of a PL time step is obtained from the first derivative in time of the vorticity maxima that was used for the detection of the PL. These maxima are based on the spatially-smoothed vorticity using a uniform filter of 60 km radius (point two of the algorithm modifications in Sec. 2.2). However, the time evolution in the vorticity is still discontinuous (Supplementary Fig. 1). Therefore a Savitzky-Golay filter (Savitzky and Golay, 1964) is applied on the time evolution in the

vorticity for the computation of the first derivative. This filter applies a least-square regression in our case of a second-order polynomial within a window of eleven vorticity time steps, or of the whole PL lifetime if this is shorter than ten hours. The **growth rate** is computed by the fraction of the vorticity tendency to the vorticity of a time step.

## 2.5 Self-organising maps (SOM)

Kohonen (2001) developed the SOM method for displaying typical patterns in high-dimensional data. The patterns, also re-

ferred to as nodes, are ordered in a 2-D array with neighbouring nodes being more similar to each other than nodes further apart in the array. Kohonen (2001) originally developed the method for artificial neural networks, but in recent years it has been extensively applied in many fields of science, including climate data analysis (e.g. Nygård et al., 2019).

We apply the package described in Wehrens et al. (2007). The size of the node array has to be subjectively chosen for the dataset at hand and is typically determined after some testing. We find an array of 3×3 nodes to be most suitable, reducing

12,695 PL-centred fields to 3×3 archetypal nodes. Larger arrays mainly display additional details of minor interest (Supplementary Fig. 2), whereas smaller arrays merge nodes that contained relevant individual information.

The SOM analysis is based on the temperature anomaly field at $850\,\text{hPa}$ of each time step $T'(x,y) = T(x,y) - \overline{T}$, with $\overline{T}$ denoting the local mean temperature within the PL-centred grid of a given time step. In this way, the fields become independent on the PL occurring in a relatively warm or cold environment, which would otherwise dominate the SOM analysis (Supplementary Fig. 3 and 4). Thereby, the intrinsic temperature structure becomes apparent. In order to illustrate the robustness of the result, we also apply the SOM algorithm to several other atmospheric fields. The SOM matrices produce similar patterns of variability when applied to the temperature anomaly field at other levels, the specific humidity anomaly, and the upper and lower-level geopotential height anomaly (Supplement Section 3). This demonstrates the generality of the results obtained from the temperature anomaly field at $850\,\text{hPa}$. Additionally, Supplement Section 5 provides evidence that the SOM algorithm is successful in detecting characteristic PL environments.

An advantage of the orientation of the PL-centred fields based on the propagation direction is a reduction of the variability in the mid-level flow, as this flow largely determines the propagation of the PLs. Therefore, the SOM matrix obtained using the mid-level geopotential height anomaly produce nodes that are fairly similar to each other (Supplementary Fig. 7), which expresses that PLs are generally characterised by a mid-level trough within a background flow in the propagation direction of the PL.

The PL time steps associated with genesis (intial), mature, and lysis (last) stages are counted for each node. The mature stage of the PL is here defined as the time step with the maximum spatially-filtered relative vorticity at $850\,\text{hPa}$, as utilised in the tracking algorithm (Supplementary Fig. 1). A PL can transition through several SOM nodes during its lifetime, which can be tracked through the SOM matrix. Evolution primarily occurs between neighbouring SOM nodes, as neighbours in the SOM matrix are most similar. Sometimes PLs transition back and forth between nodes, which indicates that the system is in a state between two nodes. We disregard this back and forth development as it does not express a clear transition of the system.

Our results are robust across multiple sensitivity tests in which subgroups of the PL track points were used: (i) PL tracks that match to a primary PL from the Rojo list, with a lifetime of at least $12\,\text{h}$, and a maximum life-time environmental near-surface wind speed exceeding $20\,\text{ms}^{-1}$, (ii) PL tracks that match in at least 5 track points with the same PL from the Rojo list within a distance of less than $75\,\text{km}$, PL track points from (iii) initial, (iv) mature, and (v) lysis time steps.

## 3 Typical polar-low configurations

### 3.1 Patterns of variability

The SOM nodes have horizontal temperature anomaly fields resembling different strength and orientation of the temperature gradient with respect to the propagation direction of the PLs (Fig. 2). Nodes in the corners are the most extreme by construction of the SOM algorithm and therefore include the strongest horizontal temperature gradients. Nodes on opposing sides of the matrix display temperature anomaly fields that are most different from each other.

PLs in SOM node 1 and 9 propagate approximately perpendicular to the horizontal temperature gradient at $850\,\text{hPa}$ with the cold side to the left and right, respectively, of the propagation direction. Thus, nodes 1 and 9 represent the classical forward and reverse-shear conditions, respectively (e.g. Forbes and Lottes, 1985; Terpstra et al., 2016). The other two nodes in the corner

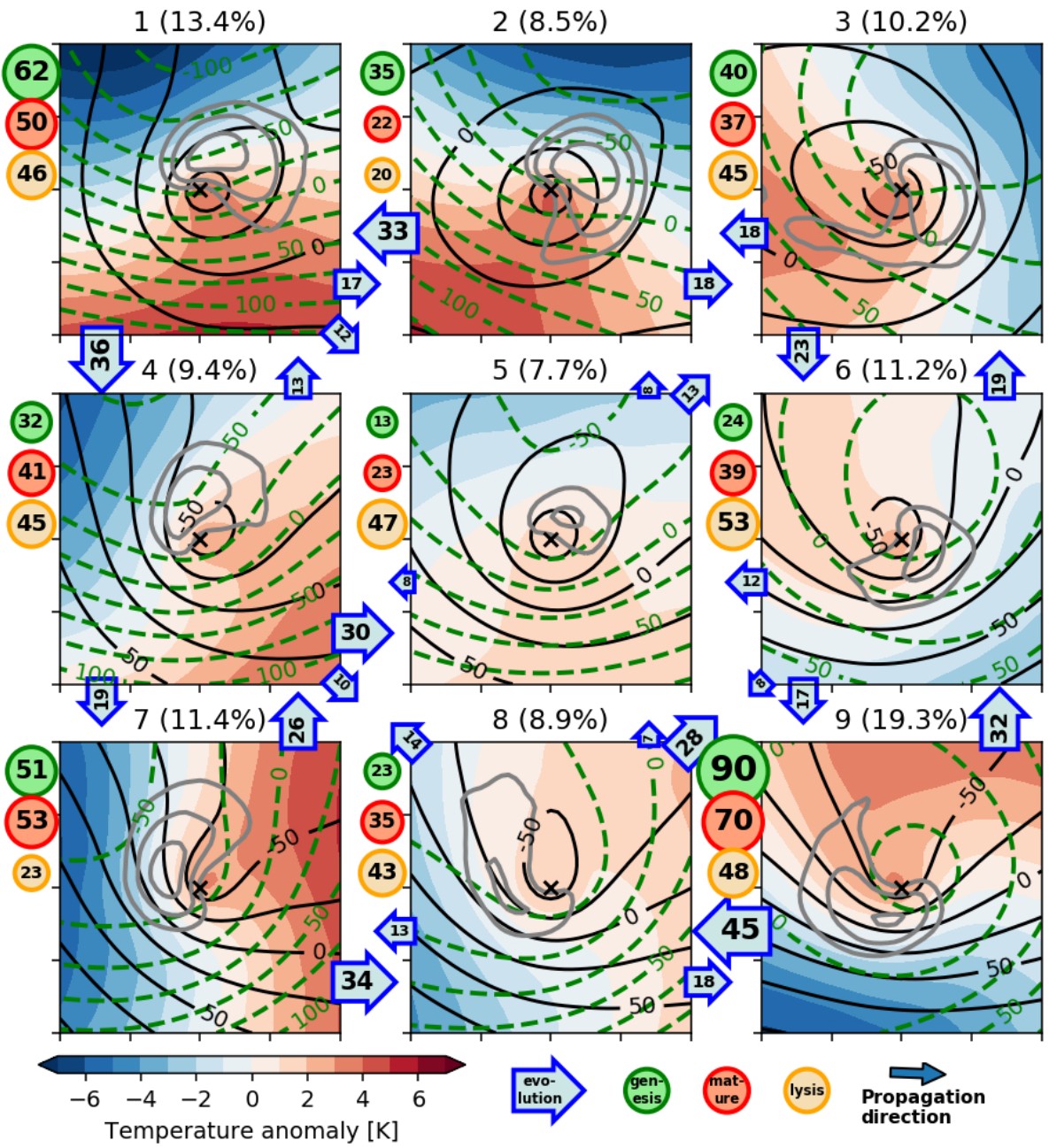

**Figure 2.** The self-organising map based on the 850 hPa temperature anomaly ($T'$) using 12,695 PL time steps in a PL-centred grid as derived in Figure 1. The black 'x' marks the PL centre, ticks on x and y-axis are spaced at 250 km. Displayed is the composite of the 850 hPa temperature anomaly (shading), 1000 hPa geopotential height anomaly (black contours, spacing 25 m), 500 hPa geopotential height anomaly (green dashed contours, spacing 25 m) and medium-level cloud-cover fraction (grey contours at 0.7, 0.8 and 0.9). The number labels the SOM nodes and the percentage of time steps represented by the respective node. Green, red, and yellow circles indicate the number of genesis, mature, and lysis stages within each node. The numbers in the arrows indicate the amount of transitions between two nodes, where numbers smaller than 5 are not displayed.

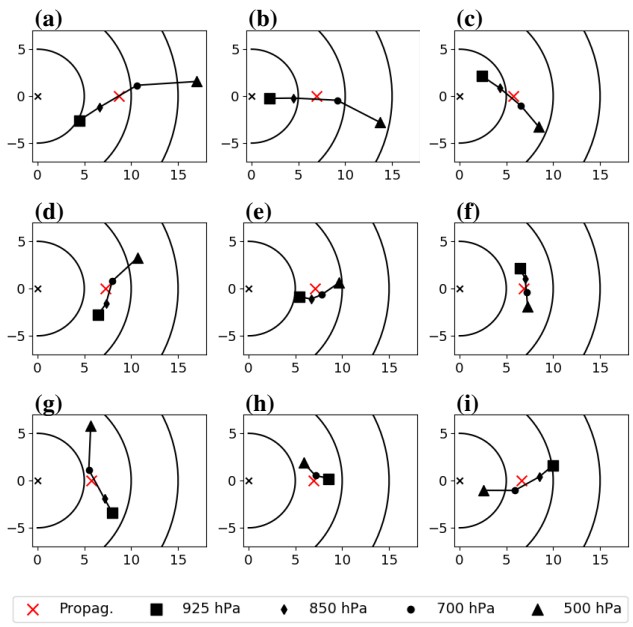

**Figure 3.** The mean hodographs associated with each SOM node displayed with the propagation direction towards the right. The square, diamond, circle, and triangle mark the SOM-mean, area-mean wind vector of the 925, 850, 700 and 500 hPa, respectively. The mean wind vectors are rotated with respect to the propagation vector of the polar low (red cross). Units on the x and y-axis are ms$^{-1}$. The origin is marked by a black "x" and black circular lines denote mean wind vectors with a strength of 5, 10 and 15 ms$^{-1}$.

of the SOM matrix, node 3 and 7, also feature a large horizontal temperature gradient. PLs in node 3 are propagating towards lower temperatures. The opposite in node 7 with PLs propagating towards higher temperatures. The remaining nodes display intermediate situations with weaker temperature gradients. Note that none of the nodes resemble axis-symmetric characteristics that would be typical for hurricane-like PLs.

     The nodes have characteristic upper (500 hPa) and lower-level (1000 hPa) flow fields (Fig. 2), where PLs in node 1 have a
closed low-level circulation and an upper-level trough located upstream (to the left of the PL centre in the depiction of Fig. 2). PLs in node 9 feature a low-level trough and a closed upper-level circulation slightly downstream. PLs in node 3 feature a weaker low-level circulation and a weak upper-level trough positioned slightly to the left of the direction of propagation. Node 7 features a short-wave low-level trough with an axis tilted from the PL centre to the left and downstream of the propagation direction, whereas the upper levels feature a trough with an axis to the left and upstream of the direction of propagation.

Contours in high values in the medium-level cloud cover associated with each node has distinct patterns (Fig. 2), which coincides with the region where the main updrafts occur (not shown). The medium-level cloud cover forms a comma shape with different orientation for each node. The cloud is typically located along the warm front on the cold side of the PL centre.

     PLs transition between SOM nodes during their life cycle (Fig. 2), which means that the orientation of the environmental flow as compared to the thermal field can change for an individual PL. Also the strength of the environmental temperature gradient

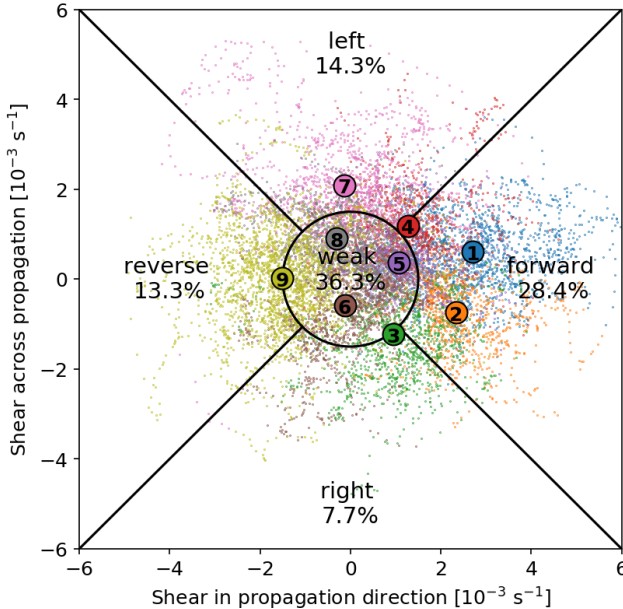

**Figure 4.** The categorisation along the vertical-shear vector (Eq. 4). The mean value of the shear vector of all time steps associated with each node is marked by a coloured circle labelled with the number of the node. The shear vector of each individual polar-low time step is displayed as a small dot in the colour that represents the SOM node of the time step. The black circle and lines divide the shear-vector space into the categories forward-shear, right-shear, reverse-shear, left-shear, and weak-shear situations. The fraction of time steps associated with each category is displayed.

varies. At genesis times (green circles in Fig. 2), PLs are most frequently associated with the SOM nodes in the corners (Fig. 2). These situations feature a strong horizontal temperature gradient. Genesis occurs most often in forward (node 1+2: 26%) and reverse shear (node 9: 24%), though also node 3 (identified as right shear in the next section: 11%) and node 7 (left shear: 14%) are common genesis situations. The SOM nodes associated with a low horizontal temperature contrast (5, 6 and 8) are predominantly lysis situations. Hence, PLs often evolve from nodes with stronger to nodes with weaker temperature gradients
in the SOM matrix.

### 3.2 Connection to vertical wind shear

In nodes 1 and 2 the area-mean wind vector increases in the vertical from around $5\,\mathrm{m\,s^{-1}}$ at $925\,\mathrm{hPa}$ [1] to around $15\,\mathrm{m\,s^{-1}}$ at $500\,\mathrm{hPa}$, both in propagation direction (Fig. 3a). This implies that the vertical-shear vector points in the direction of propagation of the system (Fig. 4), which is similar to the area-vertical-mean wind vector (Fig. 3a), hence the name forward shear. By

---

[1] Note that weak area-mean wind vectors indicate that the wind vectors cancel each other due to an almost closed cyclonic circulation near the surface (Fig. 2).

**Table 1.** Definition of the vertical-shear categories as depicted in Figure 4.

| Category | shear angle | shear strength |
|---|---|---|
| Forward | -45° to 45° | $> 1.5 \times 10^{-3}\,\mathrm{s}^{-1}$ |
| Right | 45° to 135° | $> 1.5 \times 10^{-3}\,\mathrm{s}^{-1}$ |
| Reverse | 135° to -135° | $> 1.5 \times 10^{-3}\,\mathrm{s}^{-1}$ |
| Left | -135° to -45° | $> 1.5 \times 10^{-3}\,\mathrm{s}^{-1}$ |
| Weak | all | $\leq 1.5 \times 10^{-3}\,\mathrm{s}^{-1}$ |

thermal wind balance, a forward shear is associated with the cold air being on the left of the PL as seen from the direction of propagation (Fig. 2).

Node 9 is the opposite to node 1 and 2 and features cold air to the right of the direction of propagation and thus a vertical-shear vector oriented opposite to the direction of propagation, reverse-shear conditions (Fig. 4). Reverse shear corresponds to a decrease of the strength of the mean wind vector with height, from $10\,\mathrm{ms}^{-1}$ at $925\,\mathrm{hPa}$ to $3\,\mathrm{ms}^{-1}$ at $500\,\mathrm{hPa}$ (Fig. 3i). This is consistent with Bond and Shapiro (1991) and Terpstra et al. (2016), who observed that reverse-shear systems are often accompanied by a strong low-level jet. Accordingly, reverse-shear conditions are characterised by an almost closed upper-level circulation and a strong near-surface trough (see Fig. 2).

The other two nodes with a strong vertical shear, node 3 and 7, have intermediate shear angles between forward and reverse conditions. PLs in node 3 are propagating towards colder air masses (Fig.2). The environmental flow of node 3 features warm-air advection associated with veering, a clockwise turning of the wind vector with height (Fig. 3c). The vertical wind shear is towards the right of the direction of propagation with an angle of 50±20° (Fig. 4) and hence node 3 is referred to as right-shear conditions.

Node 7 is opposite to node 3, with PLs propagating towards warmer air masses featuring environmental cold-air advection associated with backing, an anti-clockwise rotation of the wind vector with height (Fig. 3g). The vertical wind shear is towards the left of the propagation direction at an angle of -90±30° (Fig. 4). Hence, node 7 is referred to as left-shear condition, where the geostrophic flow features an upper-level trough with its axis tilted perpendicular to the left of the low-level trough axis. The same is the case for node 3, but here the axes are perpendicular in the opposite direction.

Node 4 represents an intermediate setup between node 1 and 7 with intermediate values in the shear angle (Fig. 4: -45± 20°). In the remaining nodes (5, 6 and 8) the vertical-shear strength is weak and hence the angle of the vertical shear is of less importance. The mean wind vectors of these nodes at different heights are almost uniform (Fig. 3) indicating that the flow is quasi-barotropically aligned (Fig. 2).

Given that the vertical-shear vector with respect to the propagation direction captures the different SOM nodes (Fig. 4) we suggest to use it as the key parameter to classify PLs as defined in Table 1. Sorting all PLs by their shear in propagation and cross-propagation direction, a continuous 2-dimensional parameter space emerges (Fig. 4). The thresholds are to some degree arbitrary, but variations in the thresholds were found to have no qualitative influence on the following results.

Applying the suggested sectioning of the parameter space (Fig 4), PLs in forward-shear environments generally occur more often (28.4%) than in reverse-shear situations (13.3%). Left-shear conditions (14.3%) are approximately as frequent as reverse shear conditions and right-shear situations are rather seldom (7.7%). In the following, these four categories are labelled as strong shear categories. In contrast, approximately 36.3% of the time steps have a weak shear of less than $1.5 \times 10^{-3}\,\mathrm{s}^{-1}$, which means that the wind vector is changing less than $1.5\,\mathrm{ms}^{-1}$ per km altitude.

Note that this classification is based on individual PL time steps. In this way it is considered that the environmental shear often changes during the lifetime of an individual PL. The weak-shear class is the category with most time steps, however only 38 of the 374 PLs are within this class for their whole lifetime. In contrast, 189 PLs change between strong and weak shear during their development, mainly from strong to weak shear (Fig. 2). The shear angle varies by more than 90° during the lifetime for 80 of the 336 PLs that are featuring a strong shear. Hence, the shear strength and direction varies through the lifetime of an individual PL as its ambient environment changes.

### 3.3 Characteristics of the shear categories

The thermal wind relation associates the vertical wind shear with the horizontal temperature gradient. This relation is evident for the environmental variables of the different shear categories. The strong-shear classes are exceeding a vertical wind shear of $1.5 \times 10^{-3}\,\mathrm{s}^{-1}$ and have a median horizontal temperature gradient of around $2.0\,\mathrm{K}$ per $100\,\mathrm{km}$ (Fig. 5a). PLs within these categories thus most likely intensify through baroclinc instability. In contrast, the weak-shear category has a median temperature gradient of $1.3\,\mathrm{K}$ per $100\,\mathrm{km}$ and is thus considerably less baroclinic. While hurricane-like PLs might be feasible in this more symmetric category, there is little evidence for hurricane-like intensification within this category.

Cyclogenesis of PLs in all shear categories is further supported by a low static stability ($N \approx 0.005\,\mathrm{s}^{-1}$, Fig. 5b), as a low static stability is associated with a high baroclinic growth rate (Vallis, 2017). The low static stability in PL environments is often expressed by a high temperature contrast between the sea-surface and the upper troposphere, SST-$T_{500}$ (e.g. Zappa et al., 2014; Stoll et al., 2018; Bracegirdle and Gray, 2008).

The atmosphere is slightly less stable in reverse and right-shear situations (both median of $0.0048\,\mathrm{s}^{-1}$) than in forward and left-shear situations (both median of $0.0050\,\mathrm{s}^{-1}$). This was also pointed out by Terpstra et al. (2016), who found a larger temperature contrast between the sea surface and the $500\,\mathrm{hPa}$ level for reverse compared to forward-shear conditions. It may be argued that the shear-strength threshold of $1.5 \times 10^{-3}\,\mathrm{s}^{-1}$ should be lower for reverse and right-shear categories as a weaker vertical shear can be compensated by a lower static stability. We do not adapt such an adjustment, which would increase the fraction of reverse and right-shear conditions.

Strong shear is more common in the first half of the PL lifetime (Fig. 5c) and more often associated with a positive vorticity tendency, depicting intensification (Fig. 5d). In contrast, weak shear is most common at later stages and associated with decay (70%). Even though some PL time steps in weak shear feature vortex intensification (30%), only for 6% of the weak-shear time steps the vortex is rapidly intensifying in the local-mean relative vorticity at a rate of more than $1 \times 10^{-5}\,\mathrm{s}^{-1}\,\mathrm{h}^{-1}$, whereas in strong-shear situations 22% of the time steps are associated to a vortex intensification exceeding this rate. Closer investigation

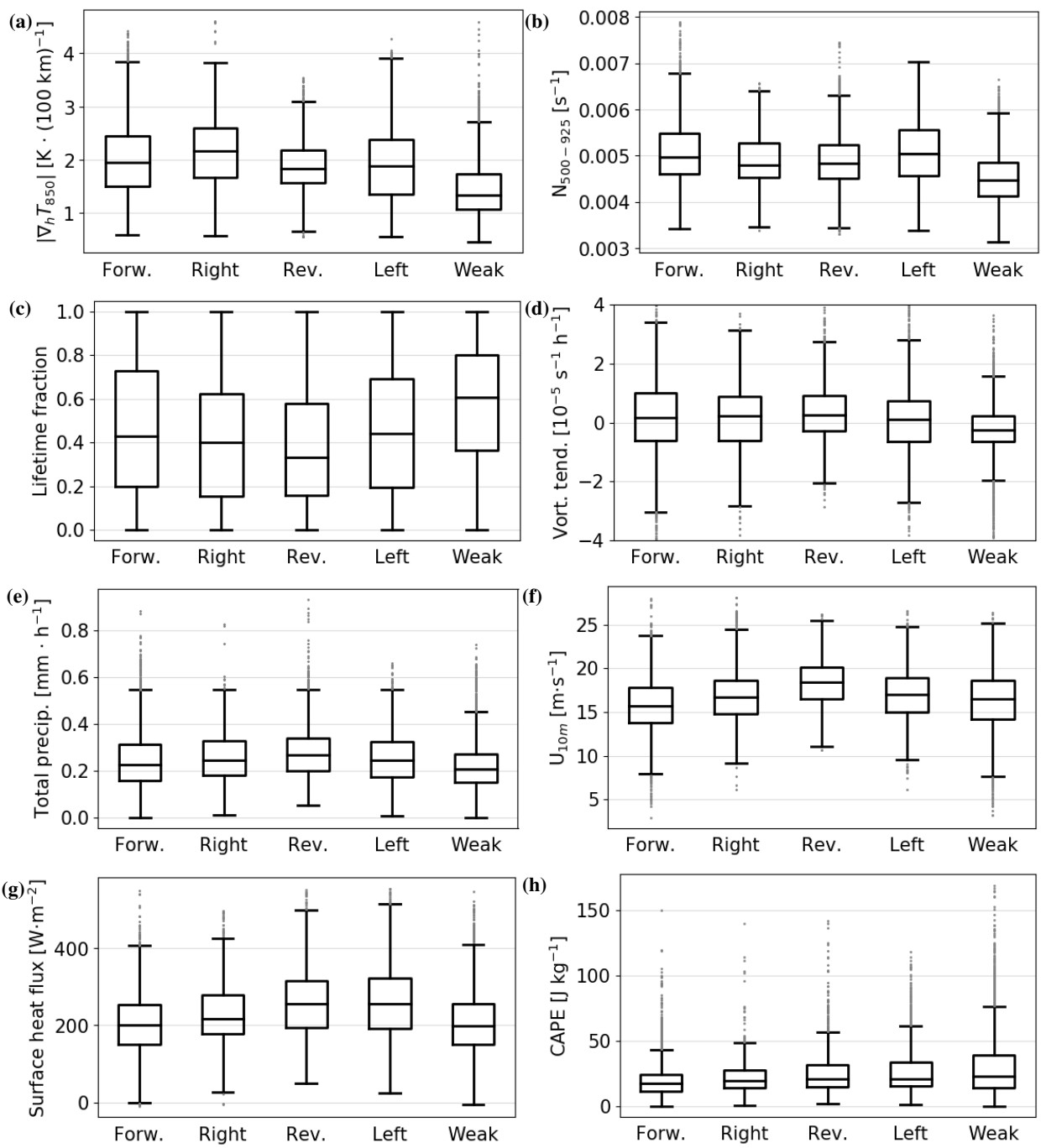

**Figure 5.** Distributions of different parameters for all time steps attributed to the shear categories introduced in Figure 4. The vorticity tendency is derived as described in Section 2.4. The lifetime fraction is given by the time step of a PL divided by its lifetime. For the other parameter, the mean value within a 250 km radius is calculated, except for the 10 m wind speed, where the maximum is applied. For the computation of the mean static stability, total precipitation, surface heat fluxes, and CAPE, grid cells covered by land or sea-ice are excluded.

reveals that these 6% of intensifying time steps within the weak-shear category feature a shear close to the threshold separating between strong and weak-shear systems.

The static stability is considerably lower in the weak-shear category (median $N = 0.0044 \, \mathrm{s}^{-1}$, Fig. 5b), which is most likely associated with this category appearing later during the PL lifetime when condensational latent-heat release has already destabilised the atmosphere.

The area-mean total precipitation rate is $0.24 \, \mathrm{mmh}^{-1}$ (median of all time steps) is rather low compared to extra-tropical cyclones. The precipitation is somewhat stronger for reverse-shear conditions (Fig. 5e, $0.27 \, \mathrm{mmh}^{-1}$), which indicates that latent heat release by condensation is most important in this class, which may compensate for a lower baroclinicity. In weak-shear condition, there is less precipitation ($0.21 \, \mathrm{mmh}^{-1}$) than for the other categories, which is consistent with this class mainly containing the decaying stages of PLs. Moreover, extreme values in precipitation are lower for weak than for strong-shear situations (Fig. 5 dots), which contradicts the idea that convective processes are more important for the weak than the strong-shear class. Furthermore, the precipitation rates appear to be insufficient to represent intensification solely through convective processes, indicating that hurricane-like dynamics are unlikely in the weak-shear class.

The median in the area-maximum near-surface wind speed of all PL time steps is $16.6 \, \mathrm{ms}^{-1}$. The near-surface winds are somewhat lower for forward (Fig. 5f, median $\approx 15.7 \, \mathrm{ms}^{-1}$) and higher for reverse-shear conditions (median $\approx 18.4 \, \mathrm{ms}^{-1}$), which is consistent with Michel et al. (2018), who found that reverse-shear polar mesoscale cyclones (PMCs) have on average a stronger lifetime-maximum near-surface wind speed ($22 \, \mathrm{ms}^{-1}$) than forward-shear PMCs ($19 \, \mathrm{ms}^{-1}$). Hence, PL detection with a criteria on the near-surface wind speed, as suggested in the definition by Rasmussen and Turner (2003) with $15 \, \mathrm{ms}^{-1}$, excludes more forward than reverse-shear systems.

In reverse-shear conditions, the environmental flow is strongest at low levels and decreases with height (3j), which is consistent with Terpstra et al. (2016). For forward-shear conditions, the environmental-mean wind vector is weak at low levels (Fig. 3a) and the near-surface wind is mainly associated with the cyclonic circulation of the PL (see also Fig. 2).

The area-mean turbulent heat fluxes at the surface are highest for left and reverse-shear conditions (median 257 and $256 \, \mathrm{Wm}^{-2}$, respectively, Fig. 5g) and slightly lower for right, forward, and weak-shear conditions (median 216, 201 and $200 \, \mathrm{Wm}^{-2}$, respectively). Higher surface turbulent heat fluxes for reverse-shear conditions were also found by Terpstra et al. (2016) and associated with the stronger near-surface winds. The higher turbulent fluxes in left-shear conditions are most likely connected to the large-scale flow being associated with cold-air advection (SOM node 7 in Fig. 2). In the weak-shear category, surface fluxes are not exceptionally high, rendering it unlikely that the WISHE mechanism is more relevant for this category than for the strong-shear categories. However, also for the strong-shear categories surface fluxes appear to have a limited direct effects on the PL intensification (Sec. 4.2), questioning the relevance of the WISHE mechanism as being of primary importance for PL development.

All shear categories feature low values in the convective available potential energy (CAPE, Fig. 5h), with median values around $20 \, \mathrm{Jkg}^{-1}$ and only a few PL time steps with CAPE above $50 \, \mathrm{Jkg}^{-1}$. This is in accordance with Linders and Saetra (2010), who found that CAPE is consumed instantaneously during PL development as it is produced. In order to be of dynamic

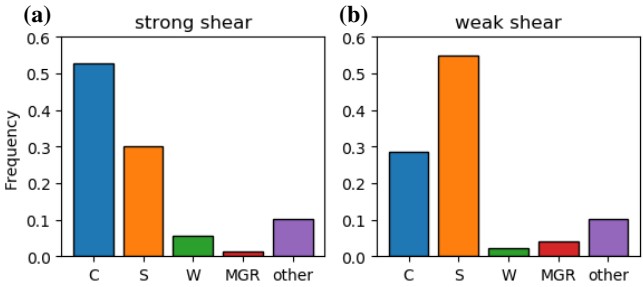

**Figure 6.** Occurrence of the cloud morphologies introduced by Rojo et al. (2019) for the time steps with a strong (left) and weak (right) vertical shear. The cloud morphologies are: C = comma shaped, S = spirali form, W = wave system, MGR = merry-go-round.

relevance, CAPE values would need to be at least one order of magnitude larger (Markowski and Richardson, 2011). Hence the CISK mechanism that relies on CAPE appears unlikely to explain intensification of PLs in the STARS dataset.

355   ### 3.4   Cloud morphology

Strong-shear conditions most commonly feature comma-shaped clouds (53% of the labelled time steps by Rojo et al. (2019)), whereas spirali-form clouds are less frequent in these categories (30%, Fig. 6). Weak-shear conditions, on the other hand, feature mainly spirali-form clouds (55%) and less frequently comma clouds (29%). This is consistent with the findings of Yanase and Niino (2007) that the cloud structure is connected to the baroclincity of the environment.

360   However, we find little evidence for axis-symmetric intensification in an environment with weak shear. Accordingly, the axis-symmetric, spirali-form system simulated by Yanase and Niino (2007) had a considerably lower growth rate than the systems within a baroclinic environment (their Fig. 3). Instead, we find that time steps in the weak-shear category resemble the occlusion stage of a baroclinic development (Fig. 2) with a quasi-barotropic alignment of the flow (Fig. 3). The spirali-form cloud signature may be explained by a baroclinic development with a warm seclusion, as suggested by the Shapiro-Keyser

365   model (Shapiro and Keyser, 1990). In later stages, this model resembles a spirali-form cloud signature. We therefore propose the warm seclusion pathway as an alternative hypothesis for the formation of PLs that appear hurricane-like.

The frequency of wave-type clouds (Rojo et al., 2015) is higher within the strong-shear (6%) than the weak-shear category (2%). Similar to comma clouds, wave-type clouds are often associated with a baroclinic development (Rasmussen and Turner, 2003). The frequency of merry-go-round systems is higher within the weak shear (4%) compared to the strong-shear categories

370   (1%). Merry-go-round are often associated with an upper-level cold cut-off low in the absence of considerable baroclinicity (Rasmussen and Turner, 2003).

### 3.5   Synoptic conditions associated with the shear categories

Within the Nordic Seas, each of the shear categories is associated with a distinct synoptic-scale situation (Fig. 7), leading to typical locations and propagation directions of PLs within each shear category. Forward-shear conditions often form south or

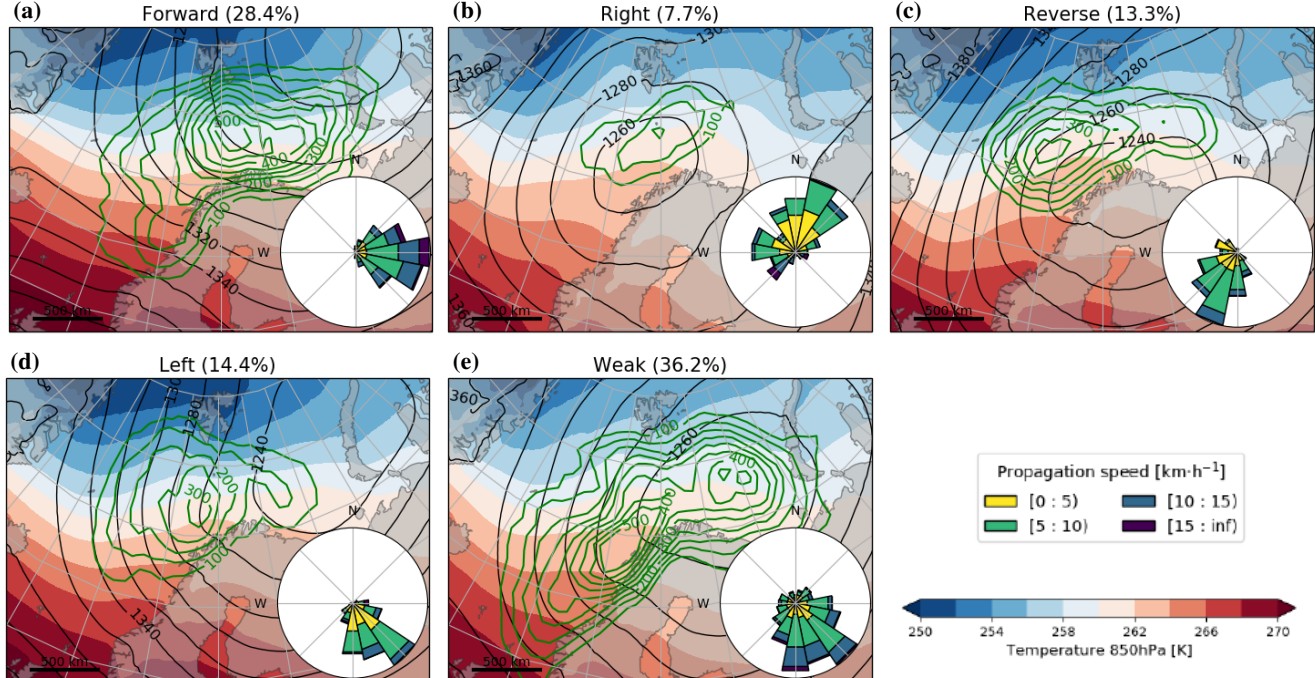

**Figure 7.** Composite maps of the 850 hPa temperature (shading) and geopotential height (black contours) associated with the polar lows within each shear category. Green contours: track densities associated with each category calculated by the number of track points within a 250 km radius. The roses depict the track distribution of the propagation direction and speed.

south-west of a synoptic-scale low-pressure system, located in the northern Barents Sea (Fig. 7a). The synoptic low causes a marine cold-air outbreak on its western side and a zonal flow further downstream on its southern side almost along the isotherms. Forward-shear PLs most frequently occur in this zonal flow and consequently propagate eastward.

The synoptic-scale situation for reverse-shear PLs mainly features a low-pressure system over northern Scandinavia leading to a cold-air outbreak from the Arctic to the Norwegian Sea, where most reverse-shear PLs develop (Fig. 7c). The flow in which reverse-shear PLs typically occur is south-westward, consistent with their direction of propagation almost along the isotherms, though with the cold side on the opposite side as seen from the direction of propagation of forward-shear PLs. The most frequent location and propagation direction of forward and reverse-shear PLs is in accordance with Terpstra et al. (2016) and Michel et al. (2018).

In left-shear conditions, a low pressure system is located over the Barents Sea causing a south to south-eastward directed cold-air outbreak across the isotherms towards a warmer environment (Fig. 7d). Hence, left-shear PLs primarily occur south of Svalbard at the leading edge of the cold-air outbreak. Right-shear PLs predominantly occur to the east and north-east of a synoptic-scale low located in the Norwegian Sea (Fig. 7b). In this situation, PLs propagate north and westward into colder air masses.

Weak-shear conditions are more variable than the other categories (Fig. 7e). PLs occur most frequently in more southerly locations near the coast of Norway and in the eastern Barents Sea, corresponding to lysis locations. A separation of the weak-shear conditions for different areas (not shown) reveals that they primarily occur downstream of one of the strong-shear categories within an area of low temperature contrast. The latter is consistent with this category mainly being associated with PLs in mature and decaying stages that originated from one of the other categories.

Multiple studies have investigated PL development associated with different weather regimes (e.g. Claud et al., 2007; Blech-schmidt, 2008; Mallet et al., 2013; Rojo et al., 2015). Comparing the typical PL propagation direction and synoptic-scale composite maps associated with the different weather regimes (e.g. Fig. 12 and 13 of Rojo et al., 2015)) and shear conditions (Fig. 7), it is apparent that forward-shear conditions somewhat resemble Scandinavian Blocking (SB), reverse shear the nega-tive phase of the North Atlantic Oscillation (NAO-), left shear the NAO+, whereas right and weak-shear situations are difficult to associate with a specific weather regime. However, composite maps of wind at 850 hPa for the Atlantic Ridge, NAO+, and NAO- featuring PLs (Rojo et al., 2015, Fig. 13a-c) are quite similar for the Norwegian and Barents Sea. Hence, the association of specific weather regimes with different shear conditions has to be considered with caution.

Furthermore, the synoptic situation for the weather regimes differ in the area of PL formation depending on whether or not PLs form (Mallet et al., 2013, Fig. 10). For example, Mallet et al. (2013) found a pattern anti-correlation of -0.4 between the normal SB pattern and the SB pattern when PLs occur. Thus, weather regimes mainly indicate whether the synoptic situation might be generally conducive for PL development, whereas the shear categories successfully identify synoptic conditions leading to different types of PL development (Fig. 7).

## 4 Intensification mechanisms

### 4.1 Baroclinic setup

The temperature as well as the upper and lower-level flow field for forward-shear PLs (Fig. 8a) resemble the structure of a smaller version of a mid-latitude baroclinic cyclone that develops along the polar front featuring a typical up-shear[2] tilt with height of the low-pressure anomaly (e.g. Dacre et al., 2012), where the trough axis of the tropopause depression is displaced against the shear vector compared to the closed surface-pressure circulation. Reverse-shear systems, are characterised by an intense low-level trough together with a tropopause trough that is centred up-shear (Fig. 8g).

Right and left-shear conditions are characterised by a closed low-level vortex or a low-level trough, respectively, and feature a tropopause depression with its trough axis located up-shear (Fig. 8d,j). Thus, all strong-shear categories feature an up-shear vertical tilt between the surface pressure anomaly and the upper-level depression, which is characteristic for baroclinic devel-opment (Holton and Hakim, 2013).

Consistent with the vertical tilt of the pressure anomaly, the low-level circulation is associated with down-gradient warm-air advection down-shear in the warm sector. For forward (reverse) shear conditions, the warm sector is ahead of (behind) the

---

[2]opposite direction to the shear vector

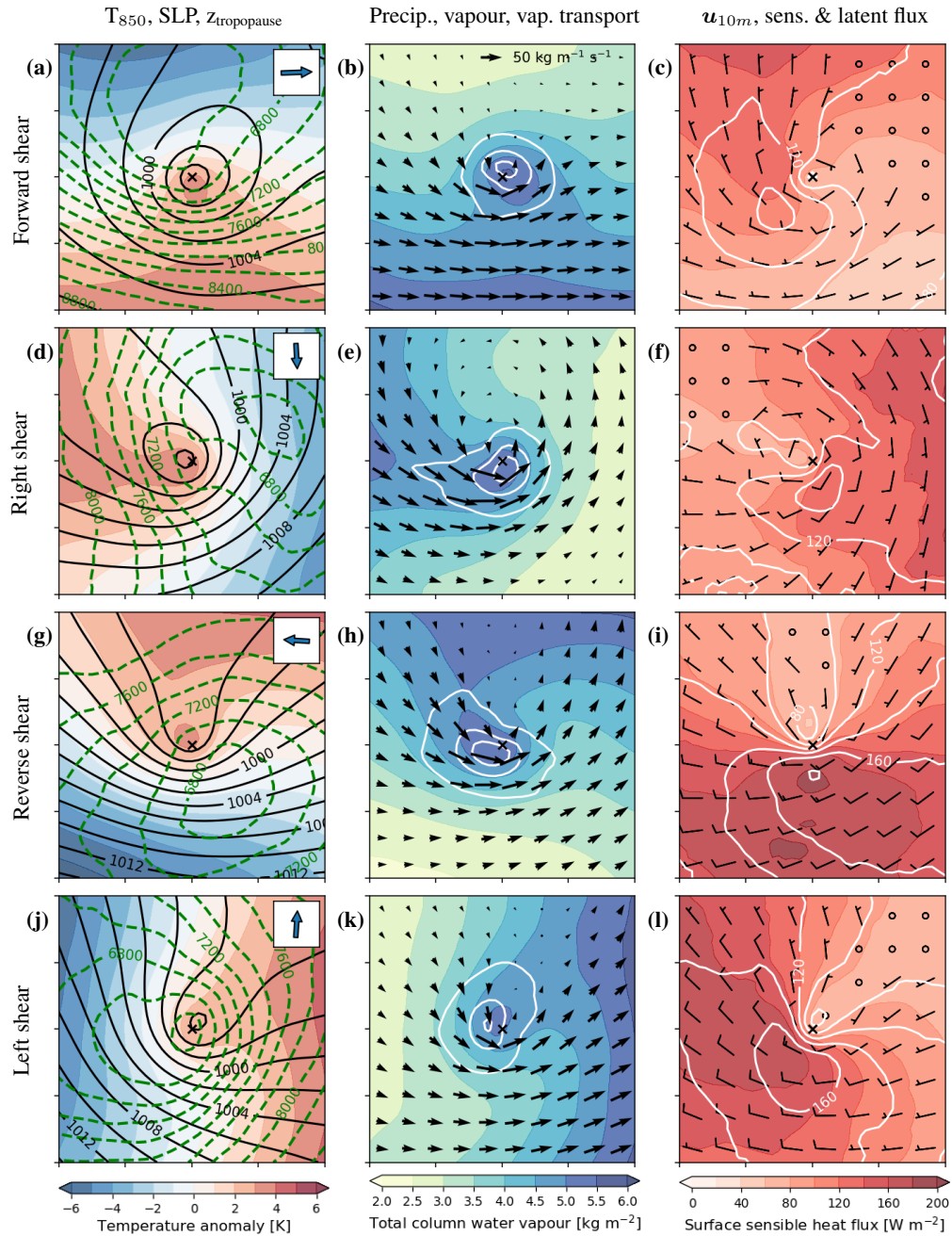

**Figure 8.** Composite maps on a PL-centred grid with propagation direction towards the right associated with four strong-shear categories. Left column: temperature anomaly at $850\,\mathrm{hPa}$ (shading), sea-level pressure (black contours, $2\,\mathrm{hPa}$ spacing), and tropopause height (green-dashed contours, spacing $200\,\mathrm{m}$). The inset shows the mean of the vertical-shear vector within the category (compare to Fig. 4). Middle column: total column water vapour (shading), total precipitation (contours, $0.2\,\mathrm{mm\,h^{-1}}$ spacing), and vertically integrated water vapour flux (arrow, reference vector at top). Right column: $10\,\mathrm{m}$ wind vectors (quivers), surface sensible heat flux (shading) and surface latent heat flux (contours, spacing $20\,\mathrm{W\,m^{-2}}$).

PL with respect to its propagation direction. For right (left) conditions, the warm sector lies to the right (left) side of the PL track. Analogously, the low-level circulation is associated with an up-gradient cold-air advection in the cold sector, which is located up-shear. Low-level temperature advection by the cyclone generates eddy available potential energy and contributes to the amplification of the PL (see first term of equation 5 in Terpstra et al. (2015)).

Downstream of the upper-level trough the flow is diverging and hereby forcing mid-level ascent (Supplementary Fig. 11),
which is co-located to the area of precipitation (second column in Fig. 8). The rising motion occurs near the surface low pressure anomaly and further intensifies the PL through vortex stretching and tilting (not shown). The interaction between the upper and lower levels is supported by a low static stability between lower and upper tropospheric levels (Fig. 5d). This suggests that the baroclinic development spans the entire depth of the troposphere and is not confined to the low levels as suggested by Mansfield (1974).

## 4.2  Diabatic contribution

Most of the precipitation occurs along the warm front in the sector left down-shear of the PL centre (Fig. 8b,e,h,k) in an area of low conditional stability ($\theta_{e,2m} - \theta_{e,500hPa} \approx -6\,K$, Supplementary Fig. 11), likely moist symmetrically neutral or slightly unstable (Kuo et al., 1991b; Markowski and Richardson, 2011). The area of precipitation is co-located with increased cloud cover featuring a comma shape (Fig. 2). The release of latent heat associated with the precipitation leads to the production
of potential vorticity underneath the level of strongest heating and hence intensifies the low-level circulation within a moist-baroclinic framework (Davis and Emanuel, 1991; Stoelinga, 1996; Kuo et al., 1991b; Balasubramanian and Yau, 1996). As the latent heat release primarily occurs in the warm sector, it further increases the horizontal temperature gradient, which contributes to the generation of eddy available potential energy (Terpstra et al., 2015).

For all shear conditions, the moisture that is converted to precipitation originates from the warm sector. In forward and
left-shear conditions, PLs are propagating towards the warm and moist sector (Fig. 8b,k), while the moisture is transported into the area of precipitation from the rear of the PL in reverse and right-shear conditions. The comma-cloud and area of main precipitation appears to be associated with the warm conveyor belt, since the trajectories that contribute to the precipitation origin in the warm sector, feature the strong ascent rates (Supplement Fig. 11) and ascent up on the warm front.

The highest sensible heat fluxes occur on the cold side of the PL (Fig. 8c,f,i,l), leading to a reduction of low-level baroclinicity
and a diabatic loss of eddy available potential energy. Latent heat fluxes are roughly co-located with the sensible heat fluxes, but occur further downstream where the air mass is already warmer and has therefore a higher capacity for holding water vapour. As the largest latent heat fluxes occur in the cold sector, the moisture released there would need to be advected around the PL to contribute to the diabatic intensification in the warm sector. Therefore, the area with maximum latent heat fluxes appear to have a limited direct effect on the intensification of the PL. The latent heat fluxes in the warm sector yield an additional source
of moisture that can more directly contribute to intensify the precipitation.

The surface heat fluxes appear to have a limited contribution, however the fluxes are important in creating an environment conducive for PL development (Kuo et al., 1991a; Haualand and Spengler, 2020). Sensible heat fluxes prior and during the PL development create an environment of low static stability, which supports the baroclinic intensification. The polar air mass

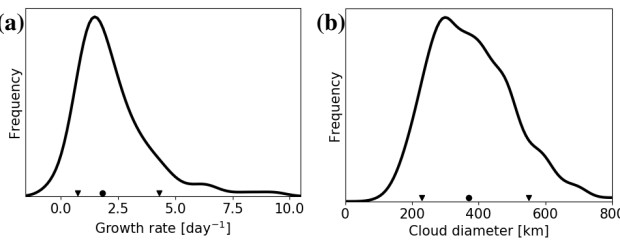

**Figure 9.** (a) Distribution of the growth-rate maximum during the PL lifetime. (b) Distribution of the cloud diameter for all PL time steps according to Rojo et al. (2019). The dot denotes the median, and the triangles the 10th and 90th percentiles of the distributions. The curves are computed with a Gaussian kernel.

**Table 2.** Approximation of values required for the determination of the growth rate, $\sigma_{max}$, and the diameter, $d_\sigma$, of the fastest growing mode by dry-baroclinic theory. For PLs, the static stability, $N$, is obtained from Figure 5, the shear strength, $\frac{u_s}{\partial z}$, from Figure 4, and the tropopause level, $H$, from Figure 8. The Coriolis parameter, $f$, is computed for $70°$ and $45°$ latitude for PLs and mid-latitude cyclones, respectively. For mid-latitude cyclones $N$, $\frac{u_s}{\partial z}$ and $H$ are approximated with the use of values from Figure 2b, 3h and 3i, respectively, of Stoll et al. (2018).

| | $N$ [s$^{-1}$] | $\frac{\partial u_s}{\partial z}$ [s$^{-1}$] | $H$ [m] | $f$ [s$^{-1}$] | $\sigma_{max}$ [day$^{-1}$] | $d_\sigma$ [km] |
|---|---|---|---|---|---|---|
| Polar lows | 0.005 | $2 \times 10^{-3}$ | 7,000 | $1.4 \times 10^{-4}$ | 1.5 | 500 |
| Mid-latitude cyclones | 0.012 | $3 \times 10^{-3}$ | 9,000 | $1.0 \times 10^{-4}$ | 0.6 | 2,400 |

in which PLs develop typically origins from sea-ice or land-covered regions (Fig. 7) and would be very dry without surface
evaporation occurring on the fetch prior the PL development. A diabatic contribution from latent heat release appears to be required in order to explain the rapid intensification of PLs (Section 4.3).

### 4.3   Scale considerations

Given the dry-baroclinic growth rate $\sigma_{max} = 0.3 \frac{f}{N} \frac{\partial u_s}{\partial z}$ and diameter of the most unstable mode $d_\sigma \approx 2 \frac{NH}{f}$ (e.g. Vallis, 2017, p.354ff), inserting typical values for PLs results in $\sigma_{max} \approx 1.5 \, \text{day}^{-1}$ and $d_\sigma \approx 500 \, \text{km}$ (Table 2), where the growth rate is
close to the observed median value of the PLs investigated in this study (1.8 day$^{-1}$, Fig.9a).

These values are quite different to typical mid-latitude cyclones, with $\sigma_{max} \approx 0.6 \, \text{day}^{-1}$ and $d_\sigma \approx 2400 \, \text{km}$ (Table 2), where the largest contribution to the faster growth and smaller scale of PLs appears to be due to the reduced static stability for PLs ($N \approx 0.005 \, \text{s}^{-1}$) compared to mid-latitude cyclones ($N \approx 0.012 \, \text{s}^{-1}$). The larger Coriolis parameter, $f$, and lower tropospheric depth, $H$, contribute only to a smaller extent to faster PL intensification and the vertical-shear, $\frac{\partial u_s}{\partial z}$, is actually weaker for PLs
compared to mid-latitude cyclones.

The estimation of the size of PLs is challenging. Often the diameter of the cloud associated with the PL is utilised for this purpose (e.g. Rojo et al., 2015), where the typical cloud diameter based on Rojo et al. (2019) is around 370 km (median, Fig. 9b). The cloud size estimated for the medium-level comma-shaped clouds of SOM 1, 3, 7 and 9 in Figure 2 is around

400 km. The vertical tilt between the upper and lower-level pressure disturbance (Fig. 8a,d,g), which in dry-baroclinic theory is a quarter of the wavelength of the fastest growing mode, is approximately 200 km, confirming the estimated diameter of around 400 km. Hence, the observed diameters are close to the theoretical estimate of 500 km.

The slight discrepancies between observation and theory are most likely attributable to latent heat release, which is observed for all shear configurations (Fig. 8b,e,h,k). The release of latent heat increases the growth rate and reduces the diameter of the fastest growing mode (Sardie and Warner, 1983; Kuo et al., 1991b; Yanase and Niino, 2007; Terpstra et al., 2015). Moist-baroclinic instability therefore appears to be the most plausible intensification mechanism for PLs, which was also proposed by Terpstra et al. (2015) and Haualand and Spengler (2020).

## 5   Discussion and conclusion

We applied the SOM algorithm to identify archetypal meteorological configurations of PL environments (Fig. 2). The different nodes in the SOM matrix display that PLs occur in environments of thermal contrast of variable strength, where the temperature gradient may take any orientation compared to the propagation direction of the system. The variability among PLs in other variables projects well on the SOM nodes (Supplement section 3).

The classification obtained from the SOM matrix can be reduced to one single variable, the vertical-shear vector with respect to the propagation direction (Fig. 4), which we use to separate PLs into five classes. We define a threshold of $1.5 \times 10^{-3} \, \mathrm{s}^{-1}$ in the vertical-shear strength to distinguish between weak-shear and strong-shear situations.

Weak-shear conditions are predominantly associated with spirali-form clouds, whereas strong-shear situations with comma-shaped clouds (Fig. 6). However, weak-shear situations occur mainly at the end of the PL lifetime and are mainly associated with decaying PL stages. In contrast, PL intensification predominantly occurs in environments with a strong vertical shear.

To identify the PL dynamics, the strong-shear situations are further separated by the vertical-shear angle into four classes. Hereby, our analysis confirms the usefulness of the classification suggested by Duncan (1978) into forward and reverse-shear PLs with the vertical-shear vector in the same or opposite direction of the PL propagation, respectively. In addition to the previously identified shear categories, we find PL configurations that feature a shear vector directed to the left or right with respect to the propagation of the PLs, which we refer to as left or right-shear conditions, respectively.

Forward-shear PLs occur predominantly in an eastward flow in the Barents Sea with cold air to the left of the direction of propagation (Fig. 7a). Reverse-shear PLs mainly develop in the Norwegian Sea in a southward flow with cold air on the right-hand side. Left-shear PLs occur at the leading edge of cold-air outbreaks and propagate towards a warmer environment, while right-shear PLs propagate towards a colder environment and occur when warmer air is advected towards a polar air mass. The shear situation of an individual PL can, however, change during its lifetime.

The baroclinic structure of the four strong shear categories features a vertical tilt between the surface and upper-level pressure anomaly against the vertical-shear vector (Fig. 8). The upper-level anomaly is captured by a tropopause depression, indicating that PLs span the entire depth of the polar troposphere. The atmospheric configuration features the classic growth through baroclinic instability, where the anomalies are organised by the vertical-shear vector. Therefore, the classification of PLs based

on their environmental thermal fields successfully reveals the dominant development mechanism. Consistent with the cloud structure, precipitation mainly develops along the warm front, which is located in the sector between the direction of the vertical-shear vector and its left side. Hence, the orientation of the comma cloud is determined by the shear vector.

The arrangement of the baroclinic structure in conjunction with the location of the latent heat release suggests a mutual interaction between the two. Latent heating enhances the baroclinicity and the diabatically-induced ascent is in phase with the baroclinically-forced adiabatic vertical motion. Thus, the effect of latent heat release is not only a linear addition to the dry-baroclinic dynamics, but also interacts directly with the adiabatic dynamics in a moist-baroclinic framework (Kuo et al., 1991b).

The surface latent heat fluxes appear to be only indirectly relevant, as the maxima in the latent heat flux are significantly displaced from the precipitation (Fig. 8). Instead, most moisture converges from the warm and moist side of the PL, as was previously observed by Terpstra et al. (2015) and Stoll et al. (2020). The direct effect of surface sensible heat fluxes would act to reduce the environmental temperature gradient and thereby most likely contributes to a dampening of the development. However, surface fluxes also shape the environment in which the PLs develop, where polar air masses would be relatively dry

without experiencing significant latent heat fluxes. The sensible heat flux reduces the static stability, which is also conducive for baroclinic development.

     Applying dry-baroclinic theory to atmospheric values for PL environments yields growth rates and diameters that are comparable to the observed values for PLs, where the discrepancies can most likely be attributed to latent heat release, which enhances growth and reduces the scale (e.g. Sardie and Warner, 1983; Terpstra et al., 2015; Haualand and Spengler, 2020). We

therefore suggest that moist-baroclinc development is the dominant mechanism leading to the intensification of the majority of PLs. The considerably higher growth rates and smaller disturbance scale of PLs as compared to mid-latitude cyclones appear to be primarily associated with a lower static stability and to a smaller extent to a higher Coriolis parameter and a lower tropopause height. The lower stability is associated with the lapse rate in cold-air outbreaks being moist adiabatic (Linders and Saetra, 2010), which, due to the low temperatures of polar air masses, is nearly equivalent to the dry adiabat.

Generally our analysis based on ERA-5 provides no evidence for the occurrence of hurricane-like intensification of PLs predominantly by convective processes within an environment of low vertical shear. This casts doubt on the PL spectrum ranging from comma-shaped, baroclinic systems to spirali-form, hurricane-like types. Instead, most PLs intensify in a baroclinic environment characterised by a strong vertical shear. However, PLs often develop a warm core (e.g. Bond and Shapiro, 1991; Nordeng and Rasmussen, 1992; Føre et al., 2011), which is typical for baroclinic development following the Shapiro-Keyser

model with a warm seclusion and a spirali-form cloud structure at the later stages of the life cycle (Shapiro and Keyser, 1990). Hence, we hypothesise that PLs with spirali-form clouds are best described as secluded cyclones, as was argued for previously (Hewson et al., 2000). To further clarify this hypothesis, studies using high-resolution datasets, such as the European regional atmospheric reanalysis CARA with a model grid-spacing of 2.5 km (Copernicus, 2020), could be used to investigate the life-cycle of PLs.

*Data availability.* The tracks of the ERA-5 matched STARS PLs are provided.

*Author contributions.* PS designed the study and performed the analysis. All authors contributed to the discussion of the methods and results. PS wrote the manuscript with contributions from all authors.

*Competing interests.* The authors declare no competing interests.

*Acknowledgements.* We thank ECMWF for providing access to data from the ERA-5 reanalysis. Parts of the data were processed at the supercomputer Stallo provided by the Norwegian Metacenter for Computational Science (NOTUR) under the project NN9348K. We were supported by Denis Sergeev, who provided access and support to the PMC-tracking algorithm and by Tiina Nygård, who shared code for the application of the SOM algorithm. Four anonymous reviewers are also thanked for their critical questions, which improved the manuscript.

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
