# Peer review of "Polar Lows – Moist Baroclinic Cyclones Developing in Four Different Vertical Wind Shear Environments"

_Weather and Climate Dynamics, 2020_

## Referee Comment (RC1) · Anonymous Referee #1 · 21 Sep 2020

This article is a significant new contribution to the knowledge of the environment conducive to PL development. With the application of a SOM technique to a large dataset extracted from ERA-5 the authors extend the separation in forward vs reverse shear for the environmental wind to include four quadrants (forward-right-reverse-left) and examine the characteristics of each, with reference to genesis and development of PL.

A rather surprising conclusion, from my point of view, is the absence of strict hurricane-like events, which would be characterized by low wind shear, warm core and symmetric structure, and surface heat fluxes actively providing energy for the development, as opposed to pre-conditioning the environment. The authors find some cases of warm core

structure in the decaying stage of PL events, which they attribute, I believe correctly, to a warm seclusion stage of a baroclinic development.

I am not familar enough with the SOM to say if the application of this technique, or of some of the operations that the authors have applied to the meteorological fields in the course of the work, may be in part responsible for this result. For example I cannot but notice that not only the mid nodes of the 3x3 map based on temperature anomaly (Fig.2 of the paper, better seen in Fig.3 of the supplement), and of the 4x5 map, fig 2 of the supplement, but also the corner node 3 of the map based on full temperature fields (fig 4 of the supplement) have a nice symmetric structure in the center. May the authors comment on the possibility or impossibility of this filtering having occurred in the analysis ?

This was my only "general" comment. I have some small observations on the text listed below:

- I am a bit confused by the numbers on pg. 4. The authors extract 556 tracks from ERA-5, and reduce them to 374 by excluding duplications and mergers. However they state that the Rojo list contains 420 PL events. So my question is are there cases in the Rojo list not found in ERA-5 ? And, secondarily, would a track search not forced to match the Rojo list have found more events in ERA-5 ?

- again pg. 4 ln 112-113. Do you mean all three of initial-middel-final times on land, or just one or two of them ?

- pg.7 ln 180 Has the overbar here the same meaning of area average as for the wind in the previous page, or an average on the square 1000x1000 km area, or is it a sort of "zonal" mean in that area ?

- pg.12 ln 264. Does this QG concept hold on these small scales ? In other words, it is not clear to me if this is a statement of principle, or it is the result of the authors study of the meteorological fields.

- several typos in the references. eg. ln 488 De Boor, De Boor, De Boor and De Boor; ln 507 Holton, American Journal of Physics (it's a book); ln 510 : what's the title ?

One final observation on the figures. I realize that it would be difficult to carry the same information in less complex figures, but the black dots in Fig.1 are only seen after enlarging the image size, and the 9 (nine!) colored dots of Fig. 4 are a challenge for those who discriminate colors poorly.
* * *

---

## Author Comment (AC1) · 14 Oct 2020

Responses to the comments of the first reviewer.

We thank the referee for the thoughtful review, which contains interesting questions and comments. In the following, we answer to each of them.

Reviewer: "I am not familiar enough with the SOM to say if the application of this technique, or of some of the operations that the authors have applied to the meteorological fields in the course of the work, may be in part responsible for this result.

For example I cannot but notice that not only the mid nodes of the 3x3 map based on temperature anomaly (Fig.2 of the paper, better seen in Fig.3 of the supplement), and of the 4x5 map, fig 2 of the supplement, but also the corner node 3 of the map based on full temperature fields (fig 4 of the supplement) have a nice symmetric structure in the center. May the authors comment on the possibility or impossibility of this filtering having occurred in the analysis?"

Response: We formulated a new section to the Supplementary material in order to answer this question:

**5 Evidence for the SOM method to detect characteristic PL environments**

Here we provide more evidence that the applied SOM method is appropriate in detecting characteristic PL environments, whereas simple composites are not sufficient.

The composite fields of meteorological variables in the PL environment may fail to represent a typical PL environment. For example, the composite of all PL time steps in the 850 hPa temperature anomaly features an almost axisymmetrical thermal structure with a warm core (Fig. 1). However, the variability in the temperature anomaly field is large (Fig. 2), especially at some distance from the PL centre. The variability exceeds the magnitude of the composite field, which indicates that PL environments may have a considerably different structure than expressed by the composite of all time steps.

The SOM algorithm is a method to detect coherent patterns of variability. The SOM matrix (e.g. Fig. 2 of paper) shows that most nodes have a different structure from the composite of all time steps. The variability in the temperature anomaly is considerably lower within each SOM node (Fig. 3) than the variability among all time steps (Fig. 2), especially at a distance of more than 100 km from the PL centre. As the variability within each SOM node is rather small and does not exceed the magnitude of

its composite field, each SOM node is considered to be representative for a typical PL environment.

In the following, we discuss why the composite of all time steps features a warm-core structure, and whether PLs can be generally considered warm-core systems. The temperature anomaly most SOM nodes is characterized by strong temperature gradients and is higher at the location of the PL centre as compared to the thermal background field (e.g. Figure 2 of the paper). It can be inferred that a mean over the thermal fields of the SOM nodes (Figure 2 of the paper) is resulting in an axisymmetric warm-core structure captured by the composite of all time steps (Fig. 1), whereas the thermal gradients of different orientation in the SOM nodes cancel out in the composite. The positive temperature anomaly at the centre within each SOM node could be attributed to the release of latent energy, or to the centre that is defined by the relative vorticity maximum, being located close to the updrafts in the warm conveyor belt, an area of low-level potential vorticity production. It is likely a combination the two reasons, possibly with other effects are contributing as well. Hence, PLs appear to be warm-core, but typically embedded in a background field of large thermal contrast.

The SOM algorithm also produces nodes (e.g. node 5 in Figure 2 of the paper) with low thermal gradients, which appears like the composite of all time steps (Fig. 1). The composite of this node features a structure that resembles an axisymmetric warm core (even more recognizable in the 4x5 SOM matrix displayed in the Appendix Figure 2). The averaging of time steps within a node might exaggerate the symmetry in the structure, by the same arguments as for the composite of all time steps. However, different from the situation with all time steps (Fig. 2), the variability is small within the symmetric SOM node (Fig. 3), which indicates that some PLs in fact have time steps with an axisymmetric warm core.

In conclusion, our method shows that axisymmetric PLs occur seldom and PL environments are instead mainly characterized by a horizontal temperature contrast.

Reviewer: "I am a bit confused by the numbers on pg. 4. The authors extract 556 tracks from ERA-5, and reduce them to 374 by excluding duplications and mergers. However they state that the Rojo list contains 420 PL events. So my question is are there cases in the Rojo list not found in ERA-5 ? And, secondarily, would a track search not forced to match the Rojo list have found more events in ERA-5"

Response: Answer to the first question: Yes, we state "374 of the 420 PL centres from the Rojo list have at least one associated PL track." So in reverse 46 PL centres from the Rojo list are not found in ERA-5 as a vorticity maxima within a distance of 150 km. However, the PL centres from the Rojo list are identified by the cloud structure, which is to some degree subjective.

Answer to the second question: This highly depends on the construction of the track search. The here applied track search detects a large amount of cyclonic systems, considerably more than in the Rojo list. However, this is not a problem for this study, as the aim of the tracking algorithm is to find the ERA-5 representation of the identified PL from the Rojo list. A study with the aim to reproduce the Rojo list as accurately as possible would need to reduce the amount of detected cyclonic systems, likely by imposing detection criteria.

Reviewer: "pg. 4 ln 112-113. Do you mean all three of initial-middel-final times on land, or just one or two of them ?"

Response: All three of them. An additional word was included: "The latter is defined as when the initial, middle, and final time step of the PL **all** occur on land."

Reviewer: pg.7 ln 180 Has the overbar here the same meaning of area average as for the wind in the previous page, or an average on the square 1000x1000 km area, or is

it a sort of "zonal" mean in that area?

Response: We changed this sentence: The SOM analysis is based on the temperature anomaly field at 850 hPa of each time step $T'(x,y) = T(x,y) - \overline{T}$, with $\overline{T}$ denoting the mean temperature within the PL-centred grid of the time step.

Reviewer: "pg.12 ln 264. Does this QG concept hold on these small scales? In other words, it is not clear to me if this is a statement of principle, or it is the result of the authors study of the meteorological fields."

Response: We tried to clarify this by replacing the sentence with the following: "The thermal wind relation associates the vertical wind shear with the horizontal temperature gradient. This relation is evident for the environmental variables of the different shear categories."

Reviewer: "several typos in the references. eg. ln 488 De Boor, De Boor, De Boor and De Boor; ln 507 Holton, American Journal of Physics (it's a book); ln 510 : what's the title?"

Response: Thanks for spotting these mistakes, we corrected them together with a few others.

Reviewer: "One final observation on the figures. I realize that it would be difficult to carry the same information in less complex figures, but the black dots in Fig.1 are only seen after enlarging the image size, and the 9 (nine!) colored dots of Fig. 4 are a challenge for those who discriminate colors poorly."

Response: Thanks for the remarks. We adapted the mentioned figures.

[Figure]

**Fig. 1.** Mean in the 850\,hPa temperature anomaly field in a PL centred perspective with propagation direction towards the right based on all time steps.

[Figure]

**Fig. 2.** Standard deviation in the 850\,hPa temperature anomaly field in a PL centred perspective with propagation direction towards the right based on all time steps.

[Figure]

**Fig. 3.** Standard deviation in the temperature anomaly at 850\,hPa for all time steps with the same SOM node. The SOM matrix is calculated as presented in Figure 2 of the paper.

---

## Referee Comment (RC2) · Anonymous Referee #2 · 21 Oct 2020

In this study, a classification of polar lows (PLs), characterized by a large variety of cloud structures, large scale environment, and intensification mechanisms is proposed It is based on PLs detected in ERA-5 during the period 1999-2019, and makes use of Self-Organizing Maps (SOM). Such a method provides different patterns of variability which can be then connected to the vertical wind shear. Five diffrent configurations are found, of which 4 correspond to a strong shear. The orientation of the shear is found to be determinant for the dynamics of the system (which justifies previous classifications in forward and reverse shear PLs). In addition, it is found that there is no evidence for the existence of hurricane-like PLs that would intensify mainly by latent heat release. Spiraliform clouds would rather correspond to secluded cyclones.

[Figure]

General comments : This is an original study on PLs for at least two reasons : so far the ability of ERA-5 to represent PLs has not been assessed, and the use of the SOM method fot displaying typical patterns in high-dimensional data, widely used by the scientific community, is certainly promising. The paper is clearly written. There are however a number of points that need to be clarified before the paper can be accepted for publication. Most of them pertain to the methodology, and have an influence on the interpretation of the results.

-Regarding the method :

a. PL detection and representation

1. How is the detection/tracking performed in ERA-5 ? In the Rojo list, there are primary PL tracks, as well as secondary PL ones (in this case, there is usually no threshhold on the associated surface wind speed, so that it is difficult to ensure that all are true PLs, according to the definition of Heinemann and Claud, 1987). Authors mention that they detect 243 of the 262 PL events of the list. In those 243 events, what is the proportion of primary PL which is detected ? This might surely affect the results (see below).

Heinemann, G., and C. Claud (1997), Report of a workshop on "Theoretical and observational studies of polar lows" of the European Geophysical Society Polar Lows Working Group, Bull. Am. Met. Soc., 78, 2643–2658.

2. A fair trajectory does not necessarily ensure that the PL is well represented, and in particular, surface wind speeds have been observed to be often under-estimated in previous reanalyses, even after downscaling (e.g. Laffineur et al, 2014). Could the authors comment on this?

3. Such methods usually detect more systems than in the reality. How many false positive PLs have been detected? Are they discarded ? In relation with the preceding point, the method of detection which has been selected is also questionable, Laffineur et al writing that "caution is required with use of the 850-hPa vorticity, which may be

indicative of troughs but not necessarily closed mesocyclonic circulations".

4. Is there a difference in the representation/detection of PL between the Norwegian and the Barents Sea, as noted by Smirnova and Golubkin, 2017 ?

This part is absolutely fundamental, since it is the basis for the results which will be obtained subsequently.

b. Regarding the use of the SOM method, there is a point that must be better justified : It seems to me that PL for which the shear situation changes during their lifetime (what is their proportion ?) should be discarded (I suspect that this would drastically reduce the size of the samples, which might be problematic). Otherwise this certainly affects the results and should prevent from drawing general conclusions. Also, would the results be modified if only primary PLs were used ?

- Concerning the interpretation of the results, one conclusion would be that there is no hurricane-like development. It might be the case in ERA-5, but this does not ensure that this is true in reality. This conclusion is too strong (at least based on the results presented here). On one side, they may not be (all) represented, and on the other hand, since these cases are probably seldom, it may well be that the method tends to smooth them. Is this method appropriate for cases that occur only occasionally ?

Minor points :

- last line of page 2 : "without an a-priori determination of a variable used for the categorisation". I don't understand this point. To me, the SOM method is applied on a single variable which has been choosen -T anomaly at 850 hPa- , it is not the algorithm which determines the variable(s) to be considered.

- Rojo et al, 2019, JGR should be quoted. (see PANGAEA site).

---

## Referee Comment (RC3) · Anonymous Referee #3 · 30 Oct 2020

In the first part of this study, a new classification of the environment of polar lows. It is based on a SOM analysis applied to a polar low dataset detected from the ERA-5 analysis. The SOM analysis reveals that the polar low environments are characterized by the vertical wind shear vector relative to the propagation direction. In the second part of this study, the development of the polar lows is discussed using composite analyses. The authors concluded that most polar lows in strong shear environment develops through moist baroclinic instability, while weak shear environment are related to mature or lysis stage of polar lows. They also concluded that spirali-form polar lows are associated with warm seclusion process, not a hurricane-like process. There is a great interest on the question the authors tried to address, and the topic of the paper

fits the scope of WCD. The SOM analysis provides an objective support for a previously identified categorization i.e. forward and reverse shear. This paper clearly shows the moist-baroclinic development of polar lows and the absence of the hurricane-like development. The results are presented clearly, and the conclusions are logically supported by the results. However, there are several points that should be clarified before the paper will be accepted. Some of them are associated with the detection methods and the rest with development mechanism of polar lows.

1. The detection method -The detection rate of the polar low depends on the thresholds used in the algorithm. Usually, weaker thresholds result in the higher detection rate, but they also cause more false detection. The false detection does not affect the polar low list in this study, because it is compiled by comparing the detected polar lows with "observed" polar lows in Rojo list. However, to evaluate the capability of ERA-5, the sensitivity to the threshold should be examined. -The authors use all timestep of detected polar lows. I think this means that polar lows with longer lifetime have larger effect on SOM analysis. Is that affects the result?

2. Development mechanism of polar lows -The authors concluded that the orientation of the vertical-shear vector for the strong shear categories determines the dynamics of the systems. However, the fundamental development mechanism is moist baroclinic processes for all strong shear categories, while there are slight differences in their environments. Please clarify what is the different dynamics between these categories. -The authors mentioned the production of the potential vorticity associated with latent heat release. If this mechanism works, polar lows tend to move the direction of the maximum precipitation, which occurs in the warm sector. This is related to the diabatic Rossby vortex mechanism indicated by Terpstra et al. (2015). However, in Fig. 8, the distribution of the precipitation is not related to the propagation of the polar lows. Do the authors conclude the DRV mechanism does not account for the development of the polar low? Please clarify.

Specific comments L. 95: Why the authors used 850 hPa vorticity? L. 116: 13221

hourly time steps for 374 PL tracks means an average lifetime of 35.4 hour. This is almost upper end of the typical lifetime of the polar low (6-36h). Is this related to the higher capability of the ERA-5, i.e. the initial stage of the polar low can be detected? L. 192: Is this mean that each PL has one time step for the mature stage and the timesteps before (after) the mature stage are categorized into genesis and lysis stage? L. 216 I think low-level trough is located slightly "down-shear" of the upper-level trough. Fig. 2: Do the amount of the transition include all timestep? If a polar low experience several transitions, are all transitions counted? Fig. 4: I recommend the same arrangement of the number in the legend as the Fig. 2. L. 345: Fig 7c -> Fig. 7d L. 347: Fig 7d -> Fig. 7b. L. 370-373: From this paragraph, I could not understand the updraft is associated with baroclinic (i.e. adiabatic) or diabatic process. Please clarify. L. 434: Could you add the information about the number of transitions between shear category like Fig. 2.

---

## Referee Comment (RC4) · Anonymous Referee #4 · 2 Nov 2020

The paper of P. Stoll and co-authors "Polar lows – moist baroclinic cyclones developing in four different vertical wind shear environments" investigates the polar lows configurations characterized by the vertical wind shear. The study is dedicated to a very sharp theme in mesoscale meteorology and climatology. A wide variety of modern methods and products are used. While studying the baroclinic polar low development, the authors claim that a hurricane-like development is not presented in the STARS dataset according to the ERA-5 data.

The manuscript presents new and valuable scientific results, well written and illustrated. At the same time, several critical issues might significantly impact the results. I list

major and minor comments below and suggest a major revision of the manuscript.

Major comment #1

Lines 138-139 – One may say that if PL radii vary from 150-600 km, in the case of small PL, you take a too large area around the vortex, which is not associated with its core. In the opposite situation, if the PL radius is 600 km, you cut the periphery, which is known to be the area of the maximum wind speed and turbulent fluxes. Depending on the addressed question, the area of 250 km around the PL center is too small (if you look at the largescale environment) or large (if you look at the processes inside the vortex itself). This might have a significant effect on the resulted composites. I would instead suggest using the estimate of PL radii presented in Rojo et al., 2015, and a buffer zone of a fixed size to cover all PL in the same way. To provide the PL on the even grid, you can normalize them before the interpolation, like somewhat presented in Rudeva and Gulev, 2011 (Rudeva, I., and S.K. Gulev Composite analysis of the North Atlantic extratropical cyclones in NCEP/NCAR reanalysis. Mon. Wea. Rev., 139, 2011, 1419-1436.)

Lines 155-157 – The same as for the previous comment. If you consider the mean wind vector in the area with a 500 km diameter around the PL center, in the case of large PLs (even 450 and 600 km in horizontal), you don't catch the environmental mean flow at all. My suggestion is to reconsider the area of the PL, at least as in the previous comment.

Figure 8 – you never describe how you compute the composites since the simple mean for even a regular grid is not enough – all PLs have different sizes. Did you do the normalization by the PL diameter? Please, see Tilinina et al., 2018, for the description of what composites are.

Major comment #2

Lines 226-227 Isn't it evident that baroclinic instability-induced vortices become ther-

[Figure]

mally homogeneous at the lysis stage? Wouldn't it be more correct to exclude lysis stages from the analysis of "archetypal meteorological conditions during the PL development" (line 62)?

Figure 2: 1. The evolution-transition blue arrows are messy and make the figure unreadable. 2. Again, the figure accounts for the same mesocyclones at different stages, which is quite confusing. I would rather plot the percentage of timesteps presented by each lifecycle stage in a particular node than the absolute value.

Minor comments

Line 33 – Talking about the hybrid nature of polar lows, I would suggest citing Terpstra et al., 2014.

Line 79 – It is better to say "ranges from 30 minutes up to 12 hours".

Line 90 – The tropopause is usually located at 300 hPa.

Lines 90-91 – 1. As far as I know, polar lows are not presented a lot poleward of 80ËŽN, so the problem of longitude convergence might not be a large one in the considered area. 2. How did you do the coarsening of data in a longitudinal direction? Did you calculate the mean of each two longitudinal points? Please, clarify it in the text.

Line 117 – How did you associate the automatically detected by the vorticity field PLs with that detected in satellite data from the Rojo list? Please, clarify it in the text.

Line 135 – why don't you just extend the ERA-5 boundaries chosen for the analysis to cover the four excluded mesocyclones? Furthermore, how did you estimate the influence of this elimination?

Line 165 – from where the 60 rm radius filter came from? Does it have any physical meaning? Please, provide an argumentation for this choice in the text of the manuscript. Why not use the Savitzky-Golay filter only?

Line 170 – "of a time step" instead of "of an time step."

Lines 192-193 – There is a more common practice to distinguish between different stages of the lifecycle of any cyclonic phenomena – normalizing the number of timesteps and letting 0-0.2 to be the genesis stage, 0.2 – 0.8 to mature stage, and 0.8 to 1 to lysis (see Simmonds, 2000; Rudeva and Gulev, 2007 for example). How can you prove your choice of the mature stage definition?

Lines 196-197 – The statements done on these lines need to be proven by some citation or description on why some signal may be simply removed from the analysis like these transition states of the system.

Line 208 – remove "is evident." Line 218 – How was the medium-level cloud cover considered?

Line 256 – I would not call the pattern depicted in fig. 4 "a spectrum." Try to recall it.

Line 266 – "most likely intensify through the baroclinic instability."

Line 281 – I would rephrase this sentence as follows: "The strong shear is more common for time steps in the first half of the PL lifetime" since your version is a bit confusing: it seems like time steps are more common, while strong shear is more common.

Lines 282-283 – the same as for the previous comment: PL time steps are not "occurring" in the philosophical sense – it may be characterized by one or another shear condition.

Lines 281-287 – please rephrase the whole paragraph. Especially at lines 284-285, where you say that time steps are intensifying. I am sure that you meant that the PLs are intensifying at these time steps.

Lines 288-289 – Did you investigate the static stability parameter separately for that 30% of PLs within the weak-shear category, which are observed at the intensification stage? This may affect your main conclusion that the "convective" PLs are not presented in the STARS dataset.

Line 312 – This statement is not fair enough. One would not expect the WISHE mechanism to be responsible for the intensification of PLs at the decaying stage (70% of time steps in this category); Secondly, it is known that the WISHE mechanism is effective at the mature stage of baroclinic MCs. For the "convective" PLs to create conditions for WISHE effect activation, it needs that the strong cyclonic circulation occurs.

Line 317 – Note that CAPE is the trigger mechanism that plays a larger role in the first half of the PL lifecycle, while the CISK is the process that occurs more in the second half (Emanuel and Rotunno, 1989), and it is better not to mix them up.

Figure 6 caption – remove "within."

Lines 336-344 – Saying "cold-air outbreak around Svalbard" (line 338) for forward-shear synoptic-scale condition and further "cold-air outbreak to the west of Svalbard" (line 341) for reverse-shear PLs, you confuse the reader because these descriptions are partly overlapping. Looking at fig.7, one may notice that the direction of the cold-air outbreak and isotherms inclination is very different for two of these cases. Thus, I would pay more attention to the text on these differences than say that both types of shear forms, particularly under the same conditions. For example, reverse-shear conditions are linked to the cold-air outbreaks over the Norwegian Sea, having the north-east direction of the mean flow, while forward-shear conditions are more about the CAO in the Barents Sea, with the mean flow directed west-north-westward.

Line 345 – "occur south of Svalbard" instead of "occur to the south of Svalbard."

Line 344 In addition to lines 336-344 – Here, left-shear conditions are typical in cold-air outbreaks with northward mean flow. This is its main difference from what we see in fig.7 for forward-shear.

Lines 348-352 – The weak-shear category is still looking not enough investigated. Fig.7e shows very clearly two types of patterns of PLs occurrence under this category. One of those is naturally the lysis near the east coast of Greenland. However, the

second, which is less numerous, should represent that 30% of generated PLs correspond to the upper-level PV anomaly-induced development of the PLs (see R&T2003, chapter 4.4, or Bracegirdle and Gray, 2007).

Line 380 – need to add a citation for the CISK mechanism realization, such as Charney and Eliassen, 1964; Rasmussen 1979; Businger and Reed, 1984; or any other that you prefer.

Figure 9 – (b) instead of (b, c)

Line 399 – check the citation.

Line 405 – "which is actually" instead of "is actually."

---

## Author Response (AR1)

We responded to Referee #1 in the open discussion.

**Response to Referee #2.**

We thank the referee for the review, which contains interesting questions and comments. In the following, we answer each of them. We provide some further evidence that support the independence of our results on tuning of the applied methodology.

Reviewer: "a. PL detection and representation
1. How is the detection/tracking performed in ERA-5?" In the Rojo list, there are primary PL tracks, as well as secondary PL ones (in this case, there is usually no threshold on the associated surface wind speed, so that it is difficult to ensure that all are true PLs, according to the definition of Heinemann and Claud, 1987). Authors mention that they detect 243 of the 262 PL events of the list. In those 243 events, what is the proportion of primary PL which is detected? This might surely affect the results (see below)."

Response: Question 1: Matching of PLs from different datasets is challenging. It is a technical procedure that relies on subjective choices. The applied tracking and matching procedure is described in the manuscript. If demanded, we can provide further clarification on specific questions.
In order to give other scientists the possibility to test and reproduce our results, we provide the obtained PL list in the supplement. This list can be investigated if doubts remain on the quality of the applied detection. We add a sentence to the end of section 2.2 to make the reader more apparent about the supplement: "The list with the obtained PL tracks is provided in the supplement."
Additionally, we add: "We compared a random subset of obtained tracks with the PLs from the Rojo list, satellite imagery and ERA-5 fields, and concluded that vast majority of the obtained tracks can be considered to be PLs."

Question 2: The primary PL is detected for 242 of the 244 [1] PL events detected from the Rojo list. Note that in total 280 of the 374 PL centres (75%) are associated to a primary PL from the Rojo list, which means that in some instances several tracked PLs are associated to the same primary PL from the Rojo list. This can occur for example if a PL matches both with a primary and a non primary PL from the Rojo list. Additionally, one Rojo PL centre, since it has sometimes observational gaps of around 10 h, appears sometimes as two individual PLs in ERA-5. In turn 94 of the 374 PLs (25%) that we investigate are associated to a secondary PL from the Rojo list.

Reviewer: "A fair trajectory does not necessarily ensure that the PL is well represented, and in particular, surface wind speeds have been observed to be often
* * *
[1]In the last version of the manuscript this number was 243. One ERA-5 PL matches to two different Rojo events and one of these events was excluded in the previous version. However this excluded Rojo event has also an associated ERA-5 track, which appears reasonable by inspection of the system.

under-estimated in previous reanalyses, even after downscaling (e.g. Laffineur et al, 2014). Could the authors comment on this?"

Response: We agree that a track match does not ensure that the PL is well represented. We therefor add a sentence to the first paragraph of section 2.2: "Further, Stoll et al. [2020] demonstrated that the ECMWF model at comparable resolution as ERA-5 reproduced the 4 dimensional structure of one PL reasonably well."
We add at the end of the paragraph: "Note that this study does not rely on the completely realistic reproduction of every PL, as we mainly focus on the PL environment, which is well captured by ERA-5."
In total, we are confident that ERA-5 is reproducing PLs. For example the lifetime-maximum near-surface wind speed of 91% of the PLs from the list exceeds the threshold of 15 m/s, the common definition threshold for PLs (Fig. 1). For the remaining 9% of the systems ERA-5 may indeed underestimate the near-surface wind speed maximum.

[Figure]

Figure 1: Frequency distribution of the 374 PLs in the lifetime-maximum area-maximum near-surface wind speed associated to the PL within a distance of 250 km from ERA-5. The distribution is estimated by a Gaussian kernel. The median, 10th and 90th percentile of the distributions are marked by a circle and triangles, respectively on the x-axis.

Reviewer: "3. Such methods usually detect more systems than in the reality. How many false positive PLs have been detected? Are they discarded ? In relation with the preceding point, the method of detection which has been selected is also questionable, Laffineur et al writing that 'caution is required with use of the 850-hPa vorticity, which may be indicative of troughs but not necessarily closed mesocyclonic circulations' "

Response: We do not agree with the interpretation that troughs are false positives. Heinemann and Claud [1997] and Rasmussen and Turner [2003] define PLs as mesoscale cyclones with some constraints. A trough is simply the superposition of a vortex (or cyclone) and the background flow. In our interpretation the PL definitions do not require that systems embedded in a strong background flow are

excluded from being PLs.

In summary, we are confident that the vast majority of the systems detected by our methodology can be classified as PLs. The near-surface wind speed distribution presented in Figure 1 provides further evidence.

Reviewer: "4. Is there a difference in the representation/detection of PL between the Norwegian and the Barents Sea, as noted by Smirnova and Golubkin, 2017?"

Response: Indeed there is a difference in accordance to Smirnova and Golubkin [2017]. We stated in line 104: "Hereby, 373 of the 420 PL centres from the Rojo list have at least one associated PL track." We remove this statement here since this comparison is done prior the exclusion due to lifetime and land. And insert the following later after line 113:

"In the Norwegian Sea 219 of 255 (86%) PL centres from the Rojo list are reproduced, whereas 129 of 165 (78%) are detected in the Barents Sea, where PLs in the Norwegian and Barents Sea are separated by the longitude of the first time step being smaller and larger than $20°$ E, respectively. A higher detection rate of STARS PLs for the Norwegian than the Barents Seas was also observed by Smirnova and Golubkin [2017]. It may be explained by STARS PLs being larger in the former than the latter ocean basin [Rojo et al., 2015] and larger systems being more likely captured by ERA-5."

Response to all points belonging to "a":

We have the impression that the reviewer indirectly asks for a sensitivity test of the robustness of the obtained results. In order to demonstrate the generality of the results, we repeat the SOM analysis on a set of systems that are with highest confidence to be classified as PLs. This set is a subset of the matched systems described in the manuscript with the following additional criteria:

- Match to a primary PL from the Rojo list. This means that systems matching only to secondary PLs from the Rojo list are excluded.

- A lifetime-maximum area-maximum near-surface wind speed that exceeds $20\,\mathrm{m/s}$, in order to ensure that the PL is having an intensity considerably larger than the threshold of $15\,\mathrm{m/s}$ that is commonly applied as intensity threshold for PLs.

- A lifetime of matched PL tracks in ERA-5 of at least $12\,\mathrm{h}$, to exclude short-lived systems that may not be well represented in ERA-5.

Of the 370 systems with 12,695 hourly time steps, the subset includes 113 PLs with 4,853 hourly time steps. The resultant SOM matrix based on the subset in the $850\,\mathrm{hPa}$ temperature anomaly (Figure 2) is similar to the SOM matrix based on the whole set (Figure 3 of Supplement). This demonstrates that the obtained results in the study are robust and not dependent on adaptations in the matching procedure. We therefore add an additional last sentence to section 2.2 of the manuscript: "The results of this study were found to be qualitatively insensitive to adaptations in the

track matching and exclusion."

[Figure]

Figure 2: As Figure 3 of the Supplement, but based on the subset of 113 PLs with 4,853 time steps that satisfy additional criteria in order to test whether the results are robust.

Reviewer: "b. Regarding the use of the SOM method, there is a point that must be better justified: It seems to me that PL for which the shear situation changes during their lifetime (what is their proportion ?) should be discarded (I suspect that this would drastically reduce the size of the samples, which might be problematic). Otherwise this certainly affects the results and should prevent from drawing general conclusions. Also, would the results be modified if only primary PLs were used?"

Response: It is a result of our study that many PLs change their shear characteristics during their lifetime, and not a weakness of the applied detection method. We further clarify this by formulating an additional paragraph at the end of Section 3.2:

"Note that this classification is based on individual PL time steps. In this way it is considered that the environmental shear often changes during the lifetime of an individual PL. The weak-shear class is the category with most time steps, however only 38 of the 374 PLs are within this class for their whole lifetime. In contrast, 189 PLs change between strong and weak shear during their development, mainly from strong to weak shear (Fig. 2). The shear angle varies by more than 90° during the lifetime for 80 of the 336 PLs that are featuring a strong shear. Hence, the shear strength and direction varies through the lifetime of an individual PL as its ambient environment changes."

Reviewer: "Concerning the interpretation of the results, one conclusion would be that there is no hurricane-like development. It might be the case in ERA-5, but this does not ensure that this is true in reality. This conclusion is too strong (at least based on the results presented here). On one side, they may not be (all) represented, and on the other hand, since these cases are probably seldom, it may well be that the method tends to smooth them. Is this method appropriate for cases that occur only occasionally?"

Response: We formulate the lack of hurricane-like PLs as an hypothesis drawn from the ERA-5 dataset. We suggest further investigation using a high-resolution dataset to test the hypothesis.

To the possibility that our methodology "smooths" hurricane-like PLs: We do not recognise outliers with hurricane-like dynamics within ERA-5 that are smoothed by our methodology. In Figure 5 of the manuscript, we present distributions in different variables and also display extreme values. We therefore formulate in the middle of section 3.3: "Moreover, extreme values in precipitation are lower for weak than for strong-shear situations (Fig. 5 dots), which contradicts the idea that convective processes are more important for the weak than the strong-shear class. Furthermore, the precipitation rates appear to be insufficient to represent intensification solely through convective processes, indicating that hurricane-like dynamics is unlikely in the weak-shear class."

Reviewer: "- last line of page 2 : "without an a-priori determination of a variable used for the categorisation". I don't understand this point. To me, the SOM method is applied on a single variable which has been choosen -T anomaly at 850 hPa- , it is not the algorithm which determines the variable(s) to be considered."

Response: We agree that this part of the sentence is misleading and therefore delete it.

Reviewer: "Rojo et al, 2019, JGR should be quoted. (see PANGAEA site)."

Response: We included the citation.

**Response to Referee #3**

We thank the third referee for the review, which contains interesting questions and comments. In the following, we answer to each of them.

Reviewer: "1. The detection method -The detection rate of the polar low depends on the thresholds used in the algorithm. Usually, weaker thresholds result in the higher detection rate, but they also cause more false detection. The false detection does not affect the polar low list in this study, because it is compiled by comparing the detected polar lows with "observed" polar lows in Rojo list. However, to evaluate the capability of ERA-5, the sensitivity to the threshold should be examined."

Response: Indeed, we could perform sensitivity tests, however an evaluation of ERA-5 is not the aim of this study. We reformulated the sentence in the abstract that specified a detection rate in ERA-5 in the previous version of the manuscript. We write in the end of Section 2.2: "Note that the detection rates depend on the applied matching criteria."

Reviewer: "-The authors use all timestep of detected polar lows. I think this means that polar lows with longer lifetime have larger effect on SOM analysis. Is that affects the result?"

Response: It is correct that polar lows with a longer lifetime have a larger effect on the analysis. We performed a SOM analysis just on the basis of initial (or mature) time steps and obtained similar SOM matrices, just with a different partitioning between the SOM nodes, or equivalently shear categories is slightly different. The weak-shear category was for example rather seldom if the SOM algorithm was applied on initial time steps only, which corresponds with results from our study (e.g. Fig. 5c). Hence, the result of the existence of different shear classes is general, but the fraction belonging to each of the shear situations is obtained on the time step level. It may well be that PLs persist in one shear situation for a longer time, a question that we did not investigate. However, a PL generally is not assigned to one shear category for the whole lifetime, which implies that partitioning into the shear categories on the system level instead of the time step level is challenging. We formulate a last paragraph to section 3.2 to make these points clearer:
"Note that this classification is based on individual PL time steps. In this way it is considered that the environmental shear often changes during the lifetime of an individual PL. The weak-shear class is the category with most time steps, however only 38 of the 374 PLs are within this class for their whole lifetime. In contrast, 189 PLs change between strong and weak shear during their development, mainly from strong to weak shear (Fig. 3). The shear angle varies by more than 90° during the lifetime for 80 of the 336 PLs that are featuring a strong shear. Hence, the shear strength and direction varies through the lifetime of an individual PL, because its

ambient environment may change. "

Reviewer: "2. Development mechanism of polar lows -The authors concluded that the orientation of the vertical-shear vector for the strong shear categories determines the dynamics of the systems. However, the fundamental development mechanism is moist baroclinic processes for all strong shear categories, while there are slight differences in their environments. Please clarify what is the different dynamics between these categories."

Response: We agree that the previous formulation was misleading. Now, we formulate l.11: "For the strong-shear categories, the shear vector organises the moist-baroclinic dynamics of the systems."

Reviewer: "-The authors mentioned the production of the potential vorticity associated with latent heat release. If this mechanism works, polar lows tend to move the direction of the maximum precipitation, which occurs in the warm sector. This is related to the diabatic Rossby vortex mechanism indicated by Terpstra et al. (2015). However, in Fig. 8, the distribution of the precipitation is not related to the propagation of the polar lows. Do the authors conclude the DRV mechanism does not account for the development of the polar low? Please clarify."

Response: The mechanism that latent heat release can provide an additional source for low-level PV in a moist-baroclinic framework without the cyclone being classified as DRV [Davis and Emanuel, 1991; Stoelinga, 1996]. It appears indeed that the propagation of PLs is mainly determined by the mid-level wind (Fig. 3 of the manuscript). The location of the precipitation as compared to the centre of the PL, which varies for the shear situations (Fig. 8 of the manuscript), has limited influence on the propagation speed or direction of the PL. Hence as the reviewer notes, from a propagation argumentation it appears as if most PLs do not follow the DRV paradigm. Also the recognised (in a composite sense in Fig. 8) interaction of an upper and lower level perturbation (in a composite sense in Fig. 8) contradicts the DRV concept. However, even though the mean situations show a baroclinic interaction, this does not exclude that some PLs have DRV characteristics, a question we do not investigate.
However, we clarify the statement in section 4.2: "The release of latent heat associated with the precipitation leads to the production of potential vorticity underneath the level of strongest heating and hence intensifies the low-level circulation within a moist-baroclinic framework [Balasubramanian and Yau, 1996; Davis and Emanuel, 1991; Kuo et al., 1991; Stoelinga, 1996]."
We further add a paragraph to the discussion and conclusion: "The arrangement of the baroclinic structure in conjunction with the location of the latent heat release suggests a mutual interaction between the two. Latent heating enhances the baroclinicity and the diabatically-induced ascent is in phase with the baroclinically-forced adiabatic vertical motion. Thus, the effect of latent heat release is not only a linear addition to the dry-baroclinic dynamics, but also interacts directly with the adiabatic dynamics in a moist-baroclinic framework [Kuo et al., 1991]."

Reviewer: "L. 95: Why the authors used 850 hPa vorticity?"

Response: We considered it beneficial to use a level from the lower part of the troposphere, as the PL definition includes significant near-surface winds, however with a some distance from the surface in order to reduce influences from the boundary. Based on Figure 3 of Yanase et al. [2004], showing relatively uniform relative vorticity values for different levels in the lower troposphere, we do not expect that the choice of the level would make a considerable difference.

Reviewer:"L. 116: 13221 hourly time steps for 374 PL tracks means an average lifetime of 35.4 hour. This is almost upper end of the typical lifetime of the polar low (6-36h). Is this related to the higher capability of the ERA-5, i.e. the initial stage of the polar low can be detected?"

Response: Indeed, we guess that the lifetime estimates from observational datasets are likely biased towards lower values, as they are based on the time difference between the first and the last observation, whereas due to large time gaps between observations of sometimes multiple hours the precise genesis and lysis time likely occurs earlier and later, respectively. On the other hand, model based climatologies are likely biased towards longer lifetimes, as the simulation of larger and longer living systems is more accurate.

Reviewer:" L. 192: Is this mean that each PL has one time step for the mature stage and the timesteps before (after) the mature stage are categorized into genesis and lysis stage?"

Response: We define the genesis and lysis stages as the first and last time time step, respectively, of a given PL. We attempt to make this more clear now: "The PL time steps associated to genesis (intial), mature, and lysis (last) stages are counted for each node."

Reviewer:" L. 216 I think low-level trough is located slightly "down-shear" of the upper-level trough?"

Response: This is correct. Since we did not introduce the perspective with respect to the shear vector at this location in the manuscript, we change the formulation to downstream: "PLs in node 9 feature a low-level trough and a closed upper-level circulation slightly downstream." We provide the up-shear perspective in Section 4.1:

Reviewer:" Fig. 2: Do the amount of the transition include all timestep? If a polar low experience several transitions, are all transitions counted?"

Response: Indeed all transitions are counted, besides the back and forth transition as formulated in the end of section 2.5.: "Sometimes PLs transition back and forth between nodes, which indicates that the system is in a state between two nodes. We

disregard this back and forth development as it does not express a clear transition of the system."

Reviewer:" Fig. 4: I recommend the same arrangement of the number in the legend as the Fig. 2.?"

Response: A good comment. Due to a comment of the first reviewer, we removed the legend and added the numbers of the nodes to the figure.

Reviewer:" L. 345: Fig 7c -> Fig. 7d?"

Response: Thanks.

Reviewer:"L. 347: Fig 7d -> Fig. 7b.?"

Response: Thanks.

Reviewer:" L. 370-373: From this paragraph, I could not understand the updraft is associated with baroclinic (i.e. adiabatic) or diabatic process. Please clarify.?"

Response: True, this is not formulated very clearly. Here, we discuss the adiabatic process and we move the part of the low stability to the diabatic section. We now state: "Downstream of the upper-level trough the flow is diverging and hereby forcing mid-level ascent (Supplementary Fig. 11), which is co-located to the area of precipitation (second column in Fig. **??**). The rising motion occurs near the surface low pressure anomaly and further intensifies the PL through vortex stretching and tilting (not shown)."

Reviewer:"L. 434: Could you add the information about the number of transitions between shear category like Fig. 2.?"

Response: A good idea, we added an additional paragraph to the end of section 3.2: "The weak-shear class is the category with most time steps, however only 38 of the 374 PLs are within this class for their whole lifetime. In contrast, 189 PLs change between strong and weak shear during their development, mainly from strong to weak shear (Fig. 2 of the manuscript). The shear angle varies by more than 90°during the lifetime for 80 of the 336 PLs that are featuring a strong shear. Hence, the shear strength and direction varies through the lifetime of an individual PL, since its ambient environment may change."

**Response to Referee #4**

We thank the fourth referee for critical questions and feedback on the manuscript, which we think led to an improved manuscript. In the following we respond to each

of the points:

Reviewer: Major comment #1. Lines 138-139 – One may say that if PL radii vary from 150-600 km, in the case of small PL, you take a too large area around the vortex, which is not associated with its core. In the opposite situation, if the PL radius is 600 km, you cut the periphery, which is known to be the area of the maximum wind speed and turbulent fluxes. Depending on the addressed question, the area of 250 km around the PL center is too small (if you look at the largescale environment) or large (if you look at the processes inside the vortex itself). This might have a significant effect on the resulted composites. I would instead suggest using the estimate of PL radii presented in Rojo et al., 2015, and a buffer zone of a fixed size to cover all PL in the same way.

Response: We agree that there is some subjectivity in the choice of the $250\,\text{km}$ radius to compute characteristics in different parameters associated to the PLs as displayed in Figure 5 of the manuscript. We therefore tested different radii for the different parameters and obtained qualitatively similar results. We added the sentence:" Presented results were found qualitatively insensitive to variations in the radius."
Calculating the parameter characteristics from the estimated radius from the Rojo list includes some challenges as well: 1. Most of the hourly time steps of the derived tracks do not have an associated Rojo track point, as the track points in the Rojo list have often time gaps of multiple hours. 2. The diameter from the cloud structure is also only a (subjective) estimate. 3. It causes problems when several PL centres are associated to one Rojo PL centre.

Reviewer: To provide the PL on the even grid, you can normalize them before the interpolation, like somewhat presented in Rudeva and Gulev, 2011 (Rudeva, I., and S.K. Gulev Composite analysis of the North Atlantic extratropical cyclones in NCEP/NCAR reanalysis. Mon. Wea. Rev., 139, 2011, 1419-1436.)

Response: This may be a good idea, but we consider the uncertainty in the scaling factor (i.e. size of the system) to be too large for performing such a normalisation. The emergent structures are clearly evident without such a scaling (Fig. 2) and it appears therefore not necessary.

Reviewer: Lines 155-157 – The same as for the previous comment. If you consider the mean wind vector in the area with a 500 km diameter around the PL center, in the case of large PLs (even 450 and 600 km in horizontal), you don't catch the environmental mean flow at all. My suggestion is to reconsider the area of the PL, at least as in the previous comment.

Response: It was written in line 157 that we use a radius of $500\,\text{km}$ and not a diameter of this distance. We are confident that the mean wind vector within a distance of $500\,\text{km}$ captures the environmental flow to a good approximation, as the influence of the PL vortex is filtered out by the mean. If the PL is to some degree point symmetric this happens independently whether the PL is fully captured within the distance. Such symmetry would also be assumed if the PL is scaled. So, we do not see a benefit of variable distances. Again, we got similar results if we chose different radii.

Reviewer: Figure 8 – you never describe how you compute the composites since the simple mean for even a regular grid is not enough – all PLs have different sizes. Did you do the normalization by the PL diameter? Please, see Tilinina et al., 2018, for the description of what composites are.

Response: Indeed, we just calculate a simple mean of the time steps associated to the category. As mentioned earlier, a normalisation has also its caveats, and the main structures become apparent without normalisation. In fact, if the size of the system would really matter and contain different structures, these structures would be revealed by the SOM algorithm, at least in a large SOM matrix as shown in the Supplementary Figure 2. In other words, the SOM matrix would show a large and a small forward-shear node, if size would matter for forward-shear systems. But it does not and hence we conclude that the size is not of primary importance.

Reviewer: Major comment 2
Lines 226-227 Isn't it evident that baroclinic instability-induced vortices become thermally homogeneous at the lysis stage?

Response: Indeed. We did not write it here since baroclinic development is not discussed before these lines.

Wouldn't it be more correct to exclude lysis stages from the analysis of "archetypal meteorological conditions during the PL development" (line 62)?

Response: We interpret the development of PLs to also include the lysis stages.

Reviewer: Figure 2: 1. The evolution-transition blue arrows are messy and make the figure unreadable. 2. Again, the figure accounts for the same mesocyclones at different stages, which is quite confusing. I would rather plot the percentage of timesteps presented by each lifecycle stage in a particular node than the absolute value.

Response: Sure, Figure 2 includes a lot of information. We think that the depiction of absolute transition captures the information best, as the evolution can be followed. For example: Node 1: 62 PLs start in this node, 17+12+36 systems leave the node, 33+13 enter the node (+3 which are not displayed since the number is insignificant as denoted in the caption), hence 46 PLs end in this state. More PLs evolve from node 1 to 4 than in the opposite direction. It can be seen that node 1 is more common for genesis than lysis situation. However, node 5, which features most often lysis situation, has almost the same amount of lysis PLs. In summary, we think that most information is captured by using absolute values instead of fractions.

Reviewer: Minor comments Line 33 – Talking about the hybrid nature of polar lows, I would suggest citing Terpstra et al., 2014.

Response: Do you mean the doctoral thesis from Terpstra?

Reviewer: Line 79 – It is better to say "ranges from 30 minutes up to 12 hours".

Response: Thanks

Reviewer: Line 90 – The tropopause is usually located at 300 hPa.

Response: The height stated here does not have any influence on the analysis. Anyway, we are more precise now in the formulation: "The reanalysis has 137 hybrid levels in the vertical, of which approximately 47 are below 400 hPa, which is the typical height of the tropopause in polar-low environments [Stoll et al., 2018]."

Reviewer: Lines 90-91 – 1. As far as I know, polar lows are not presented a lot poleward of 80N, so the problem of longitude convergence might not be a large one in the considered area. 2. How did you do the coarsening of data in a longitudinal direction? Did you calculate the mean of each two longitudinal points? Please, clarify it in the text.

Response: We do not state that longitude convergence is a problem, but we chose a coarser grid in the zonal direction to save space on our file system. At 60°latitude, a grid cell with equal lat/lon grid spacing would have a zonal length increment only half of the meridional increment. Hence, it makes more sense to use an unequal grid in the lat/lon space. However, it does not matter for our analysis, as we interpolate to a polar-low centred grid later.
The ECMWF model is spectral and the parameters are archived as spectral coefficients [2]. The requested fields are automatically converted and interpolated to a regular lat/lon grid, which the user has to define.

Reviewer: Line 117 – How did you associate the automatically detected by the vorticity field PLs with that detected in satellite data from the Rojo list? Please, clarify it in the text.

Response: We stated in line 104: "All tracks that have a distance of less than 150 km to a PL from the Rojo list for at least one time step are regarded as matches."

Reviewer: Line 135 – why don't you just extend the ERA-5 boundaries chosen for the analysis to cover the four excluded mesocyclones? Furthermore, how did you estimate the influence of this elimination?
* * *
[2]https://confluence.ecmwf.int/display/CKB/ERA5%3A+What+is+the+spatial+reference

Response: We could, but only less than 4% of the time steps that are excluded and it would require to download a considerable amount of additional data. We are confident that our results are robust that these excluded time steps would not change them.

Reviewer: Line 165 – from where the 60 rm radius filter came from? Does it have any physical meaning? Please, provide an argumentation for this choice in the text of the manuscript. Why not use the Savitzky-Golay filter only?

Response: It comes from line 99 and we specify this in the sentence of line 164. We now add a parenthesis after the sentence: "These maxima are based on the spatially-smoothed vorticity using a uniform filter of 60 km radius (point two of the algorithm modifications in Sec. 2.2)"
These spatially-filtered vorticity maxima are used to detect the PLs, as they identify locations of strong mesoscale vortexes. We think it is most consistent to use the same intensity measure as used for the identification of the systems.

Reviewer: Line 170 – "of a time step" instead of "of an time step."

Response: Thanks

Reviewer: Lines 192-193 – There is a more common practice to distinguish between different stages of the lifecycle of any cyclonic phenomena – normalizing the number of timesteps and letting 0-0.2 to be the genesis stage, 0.2 – 0.8 to mature stage, and 0.8 to 1 to lysis (see Simmonds, 2000; Rudeva and Gulev, 2007 for example). How can you prove your choice of the mature stage definition?

Response: We consider it logical to define the mature stage as the time step with highest intensity. As indicated in the response to "major point 2", by assigning only one time step per PL to each of these steps, individual PL transitions can be compared.

Reviewer: Lines 196-197 – The statements done on these lines need to be proven by some citation or description on why some signal may be simply removed from the analysis like these transition states of the system.

Response: The SOM method is a method of classifying data, where strict thresholds between the classes may not exist. Hence, if a PL is transitioning back and forth between two SOM nodes, the PL is apparently in a state between these classes, we do not think this needs any proof. Further, if we do not apply the removal of back and forth transitions the results would be similar (compare Fig. 3 to Figure 2 of the manuscript). More transitions are from strong shear to weak shear situations. However, many of the transitions from weak to strong shear are due to these back and forth transitions, hence we prefer the depiction of the manuscript.
Reviewer: Line 208 – remove "is evident."

[Figure]

Figure 3: As Figure 2 of the manuscript, but without the back and forth removal of transitions.

Response: Done.

Reviewer: Line 218 – How was the medium-level cloud cover considered?

Response: See grey lines in Figure 2 of the manuscript. We slightly change the formulation: "Contours in high values in the medium-level cloud cover associated with each node have distinct patterns (Fig. 2)"

Reviewer: Line 256 – I would not call the pattern depicted in fig. 4 "a spectrum." Try to recall it.

Response: We changed the formulation: "Sorting all PLs by their shear in propagation and cross-propagation direction, a continuous 2-dimensional parameter space emerges.

Reviewer: Line 266 – "most likely intensify through the baroclinic instability."

Response: We do not understand the purpose of this comment.

Reviewer: Line 281 – I would rephrase this sentence as follows: "The strong shear is more common for time steps in the first half of the PL lifetime" since your version is a bit confusing:it seems like time steps are more common, while strong shear is more common.

Response: Thanks. We changed accordingly: "Strong shear is more common for time steps in the first half of the PL lifetime (Fig. 5c) and more often associated with a positive vorticity tendency, depicting intensification (Fig. 5d)."

Reviewer: Lines 282-283 – the same as for the previous comment: PL time steps are not "occurring" in the philosophical sense – it may be characterized by one or another shear condition.

Response: Indeed, we also adapted this: "In contrast, weak shear is most common at later stages and associated with decay (70%)."

Reviewer: Lines 281-287 – please rephrase the whole paragraph. Especially at lines 284-285, where you say that time steps are intensifying. I am sure that you meant that the PLs are intensifying at these time steps.

Response: We followed the recommendation and rephrased the paragraph: "Strong shear is more common in the first half of the PL lifetime (Fig. 5c) and more often associated with a positive vorticity tendency, depicting intensification (Fig. 5d). In contrast, weak shear is most common at later stages and associated with decay (70%). Even though some PL time steps in weak shear feature vortex intensification (30%), only for 6% of the weak-shear time steps the vortex is rapidly intensifying in the local-mean relative vorticity at a rate of more than $1 \times 10^{-5}\,\mathrm{s}^{-1}\,\mathrm{h}^{-1}$, whereas in strong-shear situations 22% of the time steps are associated to a vortex intensification exceeding this rate."

Reviewer: Lines 288-289 – Did you investigate the static stability parameter separately for that 30% of PLs within the weak-shear category, which are observed at the intensification stage? This may affect your main conclusion that the "convective" PLs are not presented in the STARS dataset.

Response: We did an investigation now and the static stability within the weak-shear category is similar for the intensifying and the decaying time steps (median value of 0.004462 and $0.004465\,\mathrm{s}^{-1}$, respectively). As we state in the paragraph before, the 30% intensifying PLs in the weak-shear category are mainly close to the threshold to one of the strong shear categories and appear to be intensifying by baroclinic instability as well.

Reviewer: Line 312 – This statement is not fair enough. One would not expect the WISHE mechanism to be responsible for the intensification of PLs at the decaying stage (70% of time steps in this category); Secondly, it is known that the WISHE mechanism is effective at the mature stage of baroclinic MCs. For the "convective" PLs to create conditions for WISHE effect activation, it needs that the strong cyclonic circulation occurs.

Response: Could the reviewer provide some literature that demonstrate the effect of the WISHE mechanism in the mature stage of baroclinic MCs?

We did a new investigation and the surface heat fluxes within the weak-shear category is similar for the intensifying and the decaying time steps (median value of 202 and $198\,\mathrm{W/m^2}$, respectively).
We clarified our formulation: "In the weak-shear category, surface fluxes are not exceptionally high, rendering it unlikely that the WISHE mechanism is more relevant for this category than for the strong-shear categories. However, also for the strong-shear categories surface fluxes appear to have a limited direct effects on the PL intensification (Sec. 4.2), questioning the relevance of the WISHE mechanism as being of primary importance for PL development."

Reviewer: Line 317 – Note that CAPE is the trigger mechanism that plays a larger role in the first half of the PL lifecycle, while the CISK is the process that occurs more in the second half (Emanuel and Rotunno, 1989), and it is better not to mix them up.

Response: We do not understand where we mix something up. The CISK mechanism is relying on CAPE as a reservoir of potential energy, which is also formulated in the article by Emanuel and Rotunno [1989]: "The authors have recently argued against CISK as a mechanism for tropical cyclogenesis (Emanuel, 1986; Rotunno and Emanuel, 1987). In the first place, when viewed in the proper thermodynamic framework, the tropical atmosphere is very nearly neutral to deep moist convection (Betts, 1982), presumably being maintained in the convectively adjusted state by

convection itself. The reservoir of available potential energy assumed by Charney and Eliassen (1964) and Ooyama (1964) in their original concept of CISK apparently does not exist in the tropical atmosphere."

Reviewer: Figure 6 caption – remove "within."

Response: Thanks.

Reviewer: Lines 336-344 – Saying "cold-air outbreak around Svalbard" (line 338) for forward-shear synoptic-scale condition and further "cold-air outbreak to the west of Svalbard" (line 341) for reverse-shear PLs, you confuse the reader because these descriptions are partly overlapping. Looking at fig.7, one may notice that the direction of the cold-air outbreak and isotherms inclination is very different for two of these cases. Thus, I would pay more attention to the text on these differences than say that both types of shear forms, particularly under the same conditions. For example, reverse-shear conditions are linked to the cold-air outbreaks over the Norwegian Sea, having the north-east direction of the mean flow, while forward-shear conditions are more about the CAO in the Barents Sea, with the mean flow directed west-north-westward.

Response: Thanks, we changed the formulation of the first two paragraphs of this section: "Within the Nordic Seas, each of the shear categories is associated with a distinct synoptic-scale situation (Fig. 7), leading to typical locations and propagation directions of PLs within each shear category. Forward-shear conditions often form south or south-west of a synoptic-scale low-pressure system, located in the northern Barents Sea (Fig. 7a). The synoptic low causes a marine cold-air outbreak on its western side and a zonal flow further downstream on its southern side almost along the isotherms. Forward-shear PLs most frequently occur in this zonal flow and consequently propagate eastward.
The synoptic-scale situation for reverse-shear PLs mainly features a low-pressure system over northern Scandinavia leading to a cold-air outbreak from the Arctic to the Norwegian Sea, where most reverse-shear PLs develop (Fig. 7c). The flow in which reverse-shear PLs typically occur is south-westward, consistent with their direction of propagation almost along the isotherms, though with the cold side on the opposite side as seen from the direction of propagation of forward-shear PLs. The most frequent location and propagation direction of forward and reverse-shear PLs is in accordance with Terpstra et al. [2016] and Michel et al. [2018]."

Reviewer: Line 345 – "occur south of Svalbard" instead of "occur to the south of Svalbard."

Response: Thanks.

Reviewer: Line 344 In addition to lines 336-344 – Here, left-shear conditions are typical in cold-air outbreaks with northward mean flow. This is its main difference from what we see in fig.7 for forward-shear.

[Figure]

[Figure]

(a) weak shear intensifying  (b) weak shear decaying

Figure 4: As Figure 7 of the manuscript. (a) only intensifying, and (b) only decaying PLs in the weak-shear category.

Response: We reformulate this paragraph: "In left-shear conditions, a low pressure system is located over the Barents Sea causing a south to south-eastward directed cold-air outbreak across the isotherms towards a warmer environment (Fig. 7d). Hence, left-shear PLs primarily occur south of Svalbard at the leading edge of the cold-air outbreak."

Reviewer: Lines 348-352 – The weak-shear category is still looking not enough investigated. Fig.7e shows very clearly two types of patterns of PLs occurrence under this category. One of those is naturally the lysis near the east coast of Greenland. However, the second, which is less numerous, should represent that 30% of generated PLs correspond to the upper-level PV anomaly-induced development of the PLs (see RT2003, chapter 4.4, or Bracegirdle and Gray, 2007).

Response: Figure 7 display the track density of PLs in weak shear category associated to intensifying (a) and decaying (b) situations. The distribution of their locations is similar and it does not appear as if one pattern of Figure 7e of the manuscript belongs to intensifying and the other to decaying time steps.
We interpret that the "upper-level PV anomaly-induced development of the PLs" is also following a baroclinic arrangement.

Reviewer: Line 380 – need to add a citation for the CISK mechanism realization, such as Charney and Eliassen, 1964; Rasmussen 1979; Businger and Reed, 1984; or any other that you prefer.

Response: We interpret that the contribution of latent heating is best explained within a moist baroclinic framework, instead of the CISK mechanism. We formulate this more clearly and add a citation: "The release of latent heat associated with the precipitation leads to the production of potential vorticity underneath the level of strongest heating and hence intensifies the low-level circulation within a moist-baroclinic framework [Balasubramanian and Yau, 1996; Kuo et al., 1991]."
We could not find an article by Businger and Reed from the year 1984.

Reviewer: Figure 9 – (b) instead of (b, c)

Response: Thanks.

Reviewer: Line 399 – check the citation.

Response: We did.

Reviewer: Line 405 – "which is actually" instead of "is actually."

Response: We set a comma instead.

[revised manuscript text omitted]

---

## Author Response (AR2)

**Responses to the second round of comments from the second reviewer.**

We thank the referee for a second review.

Reviewer: "Regarding the PL detection and representation in ERA-5, the choice of the threshold values (tuned to have a good agreement with the Rojo list) remains a bit arbitrary. I do not understand what is meant by the authors when they mention that the results are QUALITATIVELY insensitive to the exact setting in the track matching an exclusion procedure."

Response: We agree that this formulation was a bit vague and therefore include a new paragraph at the end of the method section: "Our results are robust across multiple sensitivity tests in which subgroups of the PL track points were used: (i) PL tracks that match to a primary PL from the Rojo list, with a lifetime of at least 12 h, and a maximum life-time environmental near-surface wind speed exceeding $20\,\mathrm{ms}^{-1}$, (ii) PL tracks that match in at least 5 track points with the same PL from the Rojo list within a distance of less than 75 km, PL track points from (iii) initial, (iv) mature, and (v) lysis time steps."
Test (i) was presented in the first response. The remaining tests are provided in the following.

Reviewer: "I am also wondering whether considering all the tracks within a distance of less that 150km to a PL for at least one time step to be matches is conservative enough. Said differently, I am afraid that systems which are not PLs are included in the analysis."

Response: We provide another sensitivity test to show that the matching procedure does not influence the results. For this test we selected all PL tracks that match the same PL from the Rojo list for at least 5 different observations (from Rojo) within a radius of 75 km. 165 of the 370 PLs satisfy this matching criteria, which contain 7,630 of the original 12,695 time steps. Note, that several of the PLs from the Rojo list do not have 5 observations. The resultant SOM matrix (Fig. 1) is similar to the one from the manuscript. SOM nodes representing each of the shear categories discussed in the manuscript are recognised in this SOM matrix. Hence, the analysis described in the manuscript could be performed on basis of this subset of PL tracks.

Reviewer: "I do not agree with the explanation given to Reviewer 3 (Resp Line 116) that PLs in the Rojo list are seriously truncated because the observations which are uneven may miss the genesis and lysis periods (at least not to the point that the average duration should be 35,4 hours in ERA-5 while in Rojo et al, 2015 it is stated than most of the systems last less than 24 hours). This is a clearly a drawback in the representation in the reanalyses that should be mentioned (and this has an effect on the SOM analysis, see below)."

Response: In the previous response we discussed that a reanalysis-based climatology

[Figure]

Figure 1: As Figure 3 of the Supplement, but based on the subset of 165 PLs with 7,630 time steps that satisfy additional criteria of strict matching.

is likely biased towards longer live times, as longer living PLs are more likely well represented in the reanalysis. We also hypothesised that observational climatologies are likely underestimating the lifetime as a satellite image of the genesis and lysis time may not exist due to occasional time gaps of multiple hours between satellite images that were used for the derivation of the climatology. We now provide some more evidence: The Rojo list for example includes two PLs that have only one time step (nr. 243 and 111.b). Hence it is obvious that the full lifetime of these two PLs is not captured in the observational climatology. Further, multiple PLs in the Rojo list have less then 4 time steps. Additionally, Rojo et al. [2015] state: "In nine PL cases (out of 190 total), the formation or dissipation phases could not be observed because the temporal spacing of images did not allow tracking of such rapidly evolving systems for their entire lifespan. Despite the fact that these tracks

are probably abbreviated, we still consider them in this study."

We added an extra paragraph to the manuscript: "The resulting mean lifetime of $\sim$35 h (13,221 hourly time steps divided by 374 PLs) is slightly longer than that observed by Rojo et al. [2015], who found that two thirds of the PLs detected from satellite images lasted for less than 24 h. This difference may be explained by biases in both the model and the observational dataset. Our reanalysis-based dataset is likely biased towards longer lifetimes, as longer-living and larger PLs are better reproduced by the reanalysis. In contrast, satellite-based datasets, such as Rojo et al. [2015], are likely underestimating the lifetime based on the first (last) satellite image from which a PL may be observed after (before) the PL genesis (lysis) for two reasons: (i) Sometimes the time gap between two consecutive satellite images is multiple hours [Rojo et al., 2015], (ii) the PL is not identified due to other disturbing cloud structures [e.g. Furevik et al., 2015]."

Reviewer:"Regarding the use of the SOM method, I have 2 main points: - can the choice of a 3x3 nodes array be better justified?"

Response: We formulated in the manuscript L.192ff: "We find an array of $3\times3$ nodes to be most suitable, reducing 12,695 PL-centred fields to $3\times3$ archetypal nodes. Larger arrays mainly display additional details of minor interest (Supplementary Fig. 2), whereas smaller arrays merge nodes that contained relevant individual information." Note, that the $4\times5$ array from Supplement Figure 2 shows basically the same structures as the $3\times3$ and we could derive the shear categories from the $4\times5$ array in a similar way as described in the manuscript. Hence the size of the array has no influence on the results.

Reviewer: "- would it make sense to run the method separately for genesis, mature and lysis phases to better isolate the mechanisms (I agree with Reviewer 4 that it is not totally correct to speak of development for the lysis phase)? Also this would counterbalance (at least to some extent) the fact that PLs with a longer lifetime have a larger effect on the analysis."

Response: We performed the SOM analysis for the genesis, mature, and lysis time steps, respectively, and the results are similar to the analysis based on all time steps as utilised in the manuscript (Fig. 2). Main difference: The horizontal temperature gradients are largest at the initial and smallest at the lysis time steps. This result is included in the manuscript in Section 3.3. Further, the SOM patterns based on individual time steps during each PL (Fig. 2a-c) are locally more variable than the SOM patterns based on all time steps ((Fig. 2d), as the SOM method is based on 370 instead of 12,665 time steps.
This analysis shows again that the obtained results are robust for variations in the method. Finally, the intention here is to show the progression over different (SOM) states during the PL life time, which requires that the SOM is constructed based on all time steps.

Reviewer: "How do the results of this study relate to those which consider weather

[Figure]

Figure 2: As Figure 3 of the Supplement, but based on the (a) initial, (b) mature and (c) lysis time step of each of the 370 polar lows. For comparison (d) shows Figure 3 of the Supplement.

regimes to describe the synoptic conditions prevailing in PL cases? In my opinion, this is something that is missing and should be discussed in the paper."

Response:
Contextualising our work with previous work on the association of PL development with weather regimes could indeed be an interesting addition. We thus added the following paragraphs to the manuscript at the end of section 3.5:
"Multiple studies have investigated PL development associated with different weather regimes [e.g. Blechschmidt, 2008, Claud et al., 2007, Mallet et al., 2013, Rojo et al., 2015]. Comparing the typical PL propagation direction and synoptic-scale composite maps associated with the different weather regimes [e.g. Fig. 12 and 13 of Rojo et al., 2015]) and shear conditions (Fig. 7), it is apparent that forward-shear conditions somewhat resemble Scandinavian Blocking (SB), reverse shear the negative phase of the North Atlantic Oscillation (NAO-), left shear the NAO+, whereas right and weak-shear situations are difficult to associate with a specific weather regime. However, composite maps of wind at 850 hPa for the Atlantic Ridge, NAO+, and NAO- featuring PLs [Rojo et al., 2015, Fig. 13a-c] are quite similar for the Norwegian and Barents Sea. Hence, the association of specific weather regimes with different shear conditions has to be considered with caution.

Furthermore, the synoptic situation for the weather regimes differ in the area of PL formation depending on whether or not PLs form [Mallet et al., 2013, Fig. 10]. For example, Mallet et al. [2013] found a pattern anti-correlation of -0.4 between the normal SB pattern and the SB pattern when PLs occur. Thus, weather regimes mainly indicate whether the synoptic situation might be generally conducive for PL development, whereas the shear categories successfully identify synoptic conditions leading to different types of PL development (Fig. 7)."

Reviewer: "What is the degree of confidence which can be put in ERA-5 precipitation fields in case of PLs? Using ERA-5 precipitation to assess the relative importance of convective processes in weak/strong shear classes is really questionable."

Response: We agree that precipitation fields of ERA-5 have to be considered with some caution. However, the humidity profiles for the PL studied in Stoll et al. [2020] were of comparable quality for the ECMWF HRES model, which is quite similar to ERA-5, and a high-resolution convection-permitting model, which indicates that ERA-5 may represent moist processes reasonably well.

We present results from ERA-5 in Section 3.3 and in the conclusion we are clear about this: "Generally our analysis **based on ERA-5** provides no evidence for the occurrence of hurricane-like intensification of PLs predominantly by convective processes within an environment of low vertical shear." If the reader does not trust ERA-5, we propose the idea to test the hypothesis with a high resolution data set (see last point of this response).

Thus, the statement that convective processes appear less important for the weak than for the strong shear class makes sense within the ERA-5 model world. We see two main processes that may contribute to convection: surface sensible heat fluxes and latent heating by condensation, where the precipitation is an appropriate measure for the latter. ERA-5 captures both strong and weak-shear situations, so also weak-shear situations are produced. Both processes are, in the ERA-5 model world, of lower strength for weak than for strong shear situations.

Reviewer: "Last sentence: CARA should be described (at least in a general way), or omitted. Is it possible to add a reference?"

Response: We include a short description of CARA and improved the reference. Last sentence of the manuscript: "To further clarify this hypothesis, studies using high-resolution datasets, such as the European regional atmospheric reanalysis

CARA with a model grid-spacing of 2.5 km [Copernicus, 2020], could be used to investigate the life-cycle of PLs."

**References**

A-M Blechschmidt. A 2-year climatology of polar low events over the nordic seas from satellite remote sensing. *Geophysical Research Letters*, 35(9), 2008.

Chantal Claud, Bertrand Duchiron, and Pascal Terray. Associations between large-scale atmospheric circulation and polar low developments over the north atlantic during winter. *Journal of Geophysical Research: Atmospheres*, 112(D12), 2007.

Birgitte Rugaard Furevik, Harald Schyberg, Gunnar Noer, Frank Tveter, and Johannes Röhrs. Asar and ascat in polar low situations. *Journal of Atmospheric and Oceanic Technology*, 32(4):783–792, 2015.

Paul-Etienne Mallet, Chantal Claud, Christophe Cassou, Gunnar Noer, and Kunihiko Kodera. Polar lows over the nordic and labrador seas: Synoptic circulation patterns and associations with north atlantic-europe wintertime weather regimes. *Journal of Geophysical Research: Atmospheres*, 118(6):2455–2472, 2013.

Maxence Rojo, Chantal Claud, Paul-Etienne Mallet, Gunnar Noer, Andrew M Carleton, and Marie Vicomte. Polar low tracks over the nordic seas: a 14-winter climatic analysis. *Tellus A*, 67, 2015.

Patrick J. Stoll, Teresa M. Valkonen, Rune G. Graversen, and Gunnar Noer. A well-observed polar low analysed with a regional and a global weather-prediction model. *Quarterly Journal of the Royal Meteorological Society*, 146(729):1740–1767, 2020.

Copernicus Arctic Regional Reanalysis Service: https://climate.copernicus.eu/copernicus-arctic-regional-reanalysis-service, last access: 2020-06-11

---

## Author Response (AR3)

We are glad to hear that the manuscript got accepted.